# When Models Know More Than They Can Explain: Quantifying Knowledge Transfer in Human-AI Collaboration

**Quan Shi** [P]     **Carlos E. Jimenez** [P]     **Shunyu Yao** [OP]

**Nick Haber** [S]     **Diyi Yang** [S]     **Karthik Narasimhan** [P]

[P] Princeton Language and Intelligence

[S] Stanford University

[O] OpenAI

## Abstract

Recent advancements in AI reasoning have driven substantial improvements across diverse tasks. A critical open question is whether these improvements also yields better knowledge transfer: the ability of models to communicate reasoning in ways humans can understand, apply, and learn from. To investigate this, we introduce Knowledge Integration and Transfer Evaluation (KITE), a conceptual and experimental framework for Human-AI *knowledge transfer* capabilities and conduct the first large-scale human study (N=118) explicitly designed to measure it. In our two-phase setup, humans first *ideate* with an AI on problem-solving strategies, then independently implement solutions, isolating model explanations' influence on human understanding. Our findings reveal that although model benchmark performance correlates with collaborative outcomes, this relationship is notably inconsistent, featuring significant outliers, indicating that *knowledge transfer* requires dedicated optimization. Our analysis identifies behavioral and strategic factors mediating successful knowledge transfer. We release our code, dataset, and evaluation framework to support future work on communicatively aligned models.

## 1   Introduction

As large language models (LLMs) grow more capable, we find them quickly saturating benchmarks across reasoning-intensive domains, such as coding [6, 24, 25, 44], scientific problem-solving [40, 17, 48], and mathematics [9, 18]. A key driver, Reinforcement Learning with Verified Rewards (RLVR), has emerged as a popular post-training approach, enabling models to optimize their language outputs for high-reward reasoning in verifiable domains like math and code to achieve state-of-the-art performance and widespread industry adoption [14, 29, 52]. Yet this rapid progress hides a crucial assumption: that improvements in a model's internal reasoning naturally translate into better *knowledge transfer*, that is, a model's ability to communicate its reasoning in ways humans can understand, apply, and learn from. As we build increasingly capable reasoners, does effective knowledge transfer emerge for free, or must it be treated as a separate objective that requires dedicated evaluation and optimization?

This question has far-reaching implications. In many human-AI collaborative workflows, the goal is not merely to outsource thinking to AI, but to amplify human abilities [37, 12, 53, 15]. Without effective knowledge transfer, users may become increasingly dependent on systems they do not understand [22, 1]: a dynamic reminiscent of "manager's syndrome" [20], where individuals lose

---

[0]Correspondence to qbshi@alumni.princeton.edu. Code, data, visualizer at `kite-live.vercel.app`

39th Conference on Neural Information Processing Systems (NeurIPS 2025).

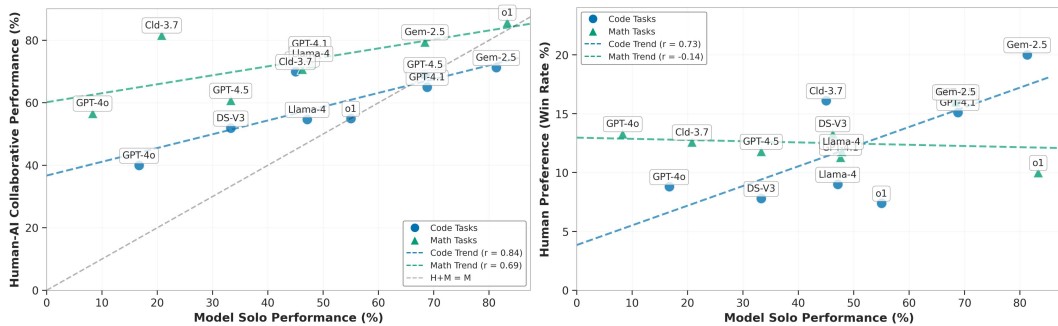

Figure 1: Left: Human-AI collaboration performance plotted against model solo performance for both code tasks (blue circles) and math tasks (green triangles). Models improve human-AI collaboration ($r = 0.84$ for code, $r = 0.69$ for math), but at a slower rate than their solo capabilities (gray line shows $y = x$). Right: Human preference rates show task-dependent correlations with model performance (positive for code tasks, $r = 0.73$; slight negative for math tasks, $r = -0.14$), revealing that user preferences vary across task domains and do not consistently align with actual performance.

technical fluency as they delegate complexity. This dynamic is further exacerbated when users cannot discern or interrogate model reasoning, leading to overreliance on systems they perceive as more intelligent, and increasing the risks of sycophantic behaviors, where models shape or reinforce user beliefs rather than supporting sound judgment. Moreover, in high-stakes settings such as medicine or legal services, the inability of models to communicate their reasoning clearly could undercut human oversight entirely [27, 21, 4]. Few works rigorously assess how well models support human understanding and enable scalable oversight, especially across latent user variables, such as differences in domain expertise, AI familiarity, or the skill gap between human and model that critically shape the success of such transfer.

To investigate this, we introduce Knowledge Integration and Transfer Evaluation (KITE), a conceptual and experimental framework that explicitly isolates and evaluates *knowledge transfer*. In our large-scale human evaluation, we recruit 118 participants with diverse levels of expertise, including a substantial proportion of domain experts (competitive programmers, math majors) who tackle challenging problems in coding and mathematics through a two-phase protocol. In the collaborative ideation phase, participants interact freely with an AI model to explore solution strategies. This phase serves as the primary opportunity for the AI to transfer knowledge to the human by explaining concepts and jointly developing solutions. In the subsequent independent implementation phase, participants attempt to implement previously discussed solutions alone, without access to the AI or any prior interaction transcripts, allowing us to isolate and measure the effectiveness of knowledge transfer. We assess outcomes using both objective metrics (solution correctness) and subjective evaluations (user rankings, perceived helpfulness, and qualitative feedback), enabling a comprehensive analysis of how well models support *knowledge transfer* across varying levels of user expertise and task difficulty. Our study is IRB approved.

As shown in Figure 1, we generally find participants demonstrated a strong ability to integrate model-generated reasoning with their own expertise. Interestingly, some models, such as Claude-3.7-Sonnet, enabled collaborative outcomes that exceeded expectations based on their solo capabilities, particularly in mathematical reasoning tasks. In contrast, higher-performing models like Gemini-2.5-Pro did not consistently yield proportionally stronger collaboration, suggesting diminishing returns in knowledge transfer as model reasoning scales. If this trend continues, as models grow more capable, their internal representations may become increasingly difficult to project in ways humans can easily understand and utilize [19].

Moreover, we find that humans' subjective preferences for models during collaboration often diverge from solo model performance, particularly in math tasks, revealing domain-specific patterns in what users value during collaboration. To probe these dynamics, we perform qualitative analyses of interaction transcripts, clustering patterns of human queries and model responses across varying user skill levels and task types. These findings surface distinct collaboration styles and success/failure modes (overreliance, representation misalignment, adaptive scaffolding...), offering a lens into the latent Human-AI interactions that govern effective knowledge transfer.

Overall, this paper aims to provide a foundation for future research on quantifying and enhancing the knowledge transfer capabilities of AI systems: particularly as models grow more intelligent and begin to develop knowledge that is increasingly inaccessible to humans. We develop a conceptual and experimental framework to isolate and quantify knowledge transfer, as well as provide insight into drivers of scaling trends between reasoning and knowledge transfer capabilities. To facilitate progress in this direction, we release our evaluation code, dataset, and filtered interaction trajectories to support future efforts in building AI systems that are more communicatively and cognitively aligned with human collaborators.

## 2   Related Work

**Human-AI Collaboration**   Research in human-AI collaboration has increasingly focused on optimizing complementary team performance and, implicitly, knowledge transfer. Studies have explored how bidirectional information exchange enhances collaborative outcomes [34, 33], examining the impact of explanations during interactions [3] and investigating how proactive AI assistants can help humans discover preferences in open-ended tasks like travel planning and data visualization [43]. Most closely related to our work, [38] evaluated the effectiveness of autocomplete suggestions and chat assistants in helping humans solve coding problems from HumanEval [6]. While these studies provide valuable insights into collaborative performance, our work extends beyond immediate task outcomes to systematically measure reasoning *transfer*.

**Code + Math Reasoning Tasks for LLMs**   Early code and math benchmarks such as HumanEval [6], MBPP [2], and GSM8k [9] focused on relatively simple problems requiring short code snippets or numerical answers. With many of these benchmarks now approaching saturation by advanced models, we deliberately selected more challenging problems from competitive programming platforms like Leetcode [24, 46] and mathematics competitions (AMC, AIME) [51]. These problems are particularly suited for our study as they primarily test reasoning abilities rather than context handling, making them ideal for measuring knowledge transfer in human-AI collaboration. This contrasts with repository-style benchmarks like SWE-Bench [25] and BigCodeBench [54], where performance is often bottlenecked by context interpretation capabilities.

**Knowledge Transfer and Education**   While limited work explicitly analyzes knowledge transfer from LLMs to humans, this shares conceptual overlap with educational applications of LLMs, where models must effectively teach reasoning to humans. Recent research has explored LLMs assisting tutors by identifying effective strategies [49], creating personalized lesson plans [26, 41, 11], providing feedback [16, 7], and functioning as specialized tutoring agents [31, 35]. However, significant challenges remain, as LLMs often underperform as teachers by leaking answers or failing to employ effective pedagogical approaches [39, 50, 13]. Our work diverges from educational applications by explicitly measuring the explanatory quality of LLM reasoning by requiring participants to independently execute discussed algorithms through mathematical calculations or code implementation, which is only possible if they truly understand the model's explanations.

## 3   KITE: Quantifying Knowledge Transfer

We first outline preliminaries for understanding *knowledge transfer* between entities during collaborative problem-solving. While we formalize knowledge regions such as $M$, $H$, and their intersections, we note that these are illustrative abstractions—difficult to precisely measure in practice, but useful for analyzing collaboration dynamics.

### 3.1   Conceptual Framework for Knowledge Transfer

We approach knowledge transfer through the lens of collective intelligence [10]: the collaborative problem-solving capability that emerges when humans and AI work together. Following [42, 28], we can represent the machine's knowledge and capabilities, or *representation space*, as $M$, and the human's as $H$; illustrated in Figure 2. This formulation yields three critical regions for our analysis:

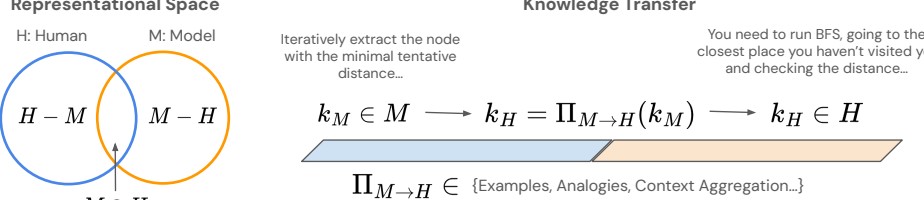

Figure 2: Model knowledge ($k_M \in M$) must be projected into a form understandable by human users ($\Pi_{M \to H}(k_M)$) in order to communicate knowledge effectively. Effective projections—via examples, analogies, or context aggregation—bridge the gap between disjoint representations.

1. **Shared Knowledge** ($M \cap H$): This intersection contains reasoning patterns, abstractions, and strategies already understood by both human and model. It forms the foundation for effective communication.

2. **AI-Exclusive Knowledge** ($M - H$): This region reflects novel reasoning, knowledge, or strategies that the model can execute but the human has not yet mastered. Transfer from this space into H is the central goal of collaborative ideation.

3. **Human-Exclusive Knowledge** ($H - M$): Reasoning held by the human but not by the model: such as intuitive understanding, prior experience or deeper domain knowledge/insight.

The success of human-AI collaboration hinges critically on accessing and transferring knowledge from the $M - H$ space into $H$, especially as humans typically maintain primary agency in collaborative tasks (e.g., deciding which strategies to pursue or when to submit solutions). However, as models become more capable, their reasoning may depend on abstractions increasingly distant from the typical human representation space. We frame this challenge in terms of projections: for each knowledge point $k_M$ in the model's space, the model must identify some *projection* $\Pi_{M \to H}(k_M)$ that translates its reasoning into a form the human can understand, internalize, and act upon. These projections can take many forms—such as providing analogies, contextualizing concepts with background knowledge, offering intermediate scaffolding, or generating concrete examples.

Importantly, this process is bidirectional. Humans also project their reasoning into the model's representation space via $\Pi_{H \to M}(k_H)$, such as using specialized prompts to elicit helpful responses. Especially in interactive settings where models are not fully autonomous, effective collaboration depends on this ongoing loop of mutual translation and aligning expressions of reasoning.

## 4 KITE: Evaluating Knowledge Transfer

Informed by the conceptualization discussed in Section 3, our two-phase setup (Figure 3) comprise a human-AI collaboration phase, and a solo human implementation phase that demands real understanding (e.g. writing code or performing calculations). Users can't simply memorize model suggestions, especially when they're incomplete or flawed; solving requires debugging, handling edge cases, and reasoning through the solution. This enables us to isolate and measure knowledge transfer from AI to humans. See Figure 17 for example problems and dataset statistics. While our setup can accommodate any reasoning problem that naturally divides into ideation and implementation phases, in this paper we focus on two domains: coding tasks from LiveCodeBench [24] and competition-level mathematics problems (AMC/AIME). These domains present consistently challenging reasoning tasks across a wide range of expertise levels, making them ideal for studying knowledge transfer.

### 4.1 Two-Phase Protocol for Isolating Knowledge Transfer

**Phase 1: Collaborative Ideation** First, participants are presented with a problem drawn from either the algorithmic coding [24] or competition mathematics [51] domains. In this phase, they engage in an open-ended dialogue with a selected LLM to explore solution strategies, exchange ideas, and scaffold their understanding without solving the problem. To preserve this ideation focus, we forbid models from generating any long-form code, pseudocode, or mathematical calculations through prompting, as well as employ a secondary checker model to withhold responses flagged to contain answers directly or indirectly (code, or mathematical calculations). Participants are also

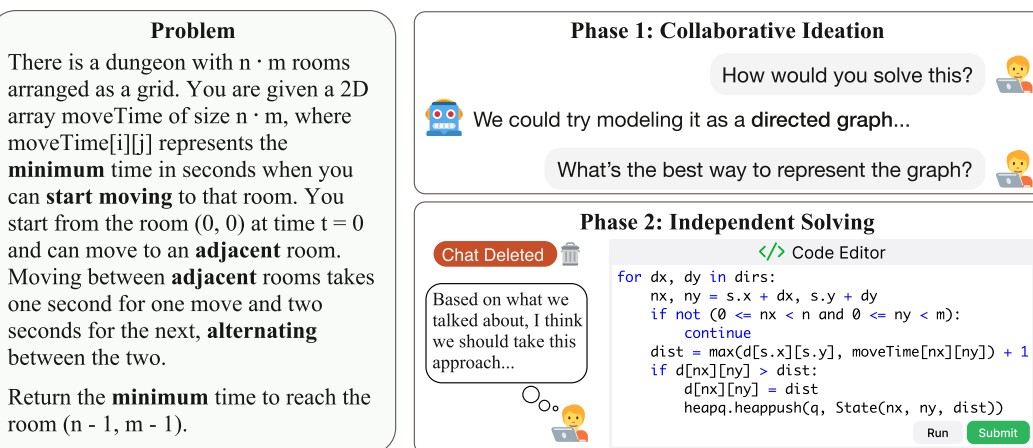

Figure 3: Two-phase evaluation framework. (1) Collaborative Ideation: Users and an AI assistant engage in open-ended discussion to explore problem-solving strategies. (2) Independent Solving: Users then implement a solution independently, without further assistance. This design leverages the nature of coding and math tasks—where successful implementation demands deep understanding, not rote recall—to isolate and measure genuine knowledge transfer.

not allowed to take any notes to log model insights. We additionally perform post-hoc filtering to remove user interaction data where models emit forbidden content. This ensures that any knowledge transferred takes the form of conceptual reasoning or strategy, rather than memorization of content that can be *directly* used to assemble the final solution.

**Phase 2: Independent Solving**    After the ideation phase concludes, the LM interface and conversation history are *no longer accessible*. Participants are tasked with solving the exact same problem on their own, without model assistance. In coding, participants must write and submit correct implementations that pass all test cases, given 10 code submission attempts. In math, participants must carry out precise multi-step calculations to arrive at a final answer, given 5 answer submission attempts. By requiring participants to independently execute a solution, Phase 2 becomes a direct and rigorous test of whether they have absorbed and retained reasoning introduced in Phase 1. Successful completion indicates that knowledge previously exclusive to the model ($k_M \in M - H$) has been projected into and re-applied by the human ($\Pi_{M \to H}(k_M) \in H$).

## 4.2    Modeling and Calibrating Skill Hierarchies

Collaboration becomes meaningful only when the task challenges the human's independent capabilities. If the human can already easily solve the problem alone, model assistance becomes redundant: there is no opportunity for knowledge transfer, no dependency, and thus no true collaboration. This necessitates the calibration of *skill hierarchies*: the relative proficiencies of the human, the model, and the task. We accomplish this by assigning standardized skill ratings (elo) to each of the three entities in the problem-solving process.

**Skill Estimation**    *Task difficulty* is determined using externally validated Elo ratings: public Leet-Code ratings for programming tasks[1] and competition-derived estimates for AMC/AIME math problems[2]. *Human skill* is estimated through a two-step process: participants self-report their experience level, then complete 5 adaptively selected tasks with difficulty adjusted based on performance. Their Elo rating is updated using surprise-conditioned rules (Appendix C.8) [8], yielding an empirically grounded skill estimate. *Model skill* is measured by zero-shot performance—each model attempts each task three times, and a task is considered solvable if at least one completion is correct. To compare human and model skill fairly, we contrast each human's final Elo with the average difficulty (Elo) of the top 25% of problems solved by the model, avoiding bias from models attempting all tasks regardless of difficulty.

---

[1] https://github.com/zerotrac/leetcode_problem_rating
[2] https://artofproblemsolving.com/wiki/index.php/AoPS_Wiki:Competition_ratings

**Test-Time Pairing**   During the test phase, each participant is required to solve between 3 and 15 problems. They may choose to solve any number of problems within this range and are allowed to work at their own pace, including non-contiguous problem-solving sessions. For each problem attempted, the participant is paired with one of eight held-out LLMs, sampled uniformly at random without replacement. Once all models have been encountered, the sampling process resets. Each task is selected to fall within a calibrated difficulty band slightly above the participant's demonstrated skill level, ensuring it is challenging yet tractable with model assistance. Specifically, tasks are drawn from a fixed Elo margin relative to the participant's current rating: $[t + 200, t + 400]$ for coding problems and $[t + 0.75, t + 1.25]$ for math problems. This design encourages meaningful collaboration with the model, avoiding both trivial and overly difficult cases. While task completion time is recorded, no time limits are imposed: we record this metric in Appendix B.

### 4.3   Experimental Controls and Evaluation Strategy

**Evaluation and Success Metrics**   Evaluating human-AI collaboration is challenging due to the subjective and noisy nature of human preferences. We use both subjective and objective metrics. Subjectively, after each task, participants rank the last four models they interacted with from most to least preferred; we apply the Bradley-Terry model to convert these rankings into win rates reflecting relative preference (full algorithm in Appendix C.7). We also provide separated win rates based on the relative ordering of Human, Model, and Task Skill ratings at the time of interaction. Specifically, we report results for the three possible configurations: Human > Task > Model (HTM), Human > Model > Task (HMT), and Model > Human > Task (MHT). On the objective side, we assess transfer by comparing the percentage of problems solved through human-model collaboration to the model's solo performance on the same problems. For coding tasks, correctness requires passing all associated test cases; for math, we require an exact answer match.

**Incentives and Motivation**   A common confounding factor in human-AI interaction studies is participant motivation [45, 36]: specifically, it is imperative that users are genuinely trying to learn from the model to improve their own performance. To mitigate this, first, we provide monetary incentives: participants receive 1.2x - 1.5× their base compensation of $25/hr for correctly answering a question, depending on difficulty. Second, most of our participants are actively preparing for career interviews that require proficiency in the task domains we test—e.g., competition math for finance related roles, and LeetCode-style problems for software engineering positions. This creates an added layer of intrinsic motivation: participants have a personal stake in learning from the model outputs and in providing thoughtful, honest feedback.

**Participant Selection**   We recruited participants through university-wide email advertisements and word of mouth. Interested individuals completed an initial survey, after which we filtered for a diverse sample across academic background, domain expertise, and AI/LLM familiarity to reflect a broad population representative of both technical and non-technical users. Our final cohort comprised 118 participants from 11 institutions, spanning a wide range of majors, including Computer Science ($N = 49$), Electrical Engineering, Mathematics, Neuroscience, and various STEM disciplines. Most were in their first ($N = 38$) or second ($N = 36$) year of study, though all undergraduate levels were represented. A full demographic account can be found in Appendix A.

**Model Selection**   We evaluate eight LLMs of different sizes and abilities: GPT-4.1, GPT-4o [23], GPT-4.5-preview, Gemini-2.5-Pro, DeepSeek-V3 [30], Claude-3.7-Sonnet, LLaMA-4-Maverick, and o1. These models were selected based on strong leaderboard performance on ChatArena [8] and widespread usage in interactive evaluation settings. Notably, DeepSeek-R1 [14] was considered but excluded due to availability and latency constraints. To assess natural explanation behaviors, we evaluate models in a zero-shot setting without prompt optimization or fine-tuning for explanatory quality, with temperature 0.7 when possible. This design choice avoids confounding effects of tailored prompts and better reflects how users commonly interact with models out-of-the-box.

## 5   Results

The main quantitative results of the study can be found in Figure 1 and 2. In total, we obtained 578 problem solving trajectories, with each participant completing an average of 4.90 problems. We

| Model | Code (N=300) | | | | Math (N=278) | | | |
|---|---|---|---|---|---|---|---|---|
| | HTM | HMT | MHT | Total | HTM | HMT | MHT | Total |
| GPT-4.1 | $18.6_{\pm 7.2}$ | $\mathbf{15.3_{\pm 2.0}}$ | $8.7_{\pm 4.2}$ | $15.1_{\pm 7.0}$ | $8.2_{\pm 2.9}$ | $10.7_{\pm 6.3}$ | $16.1_{\pm 3.5}$ | $11.3_{\pm 2.0}$ |
| GPT-4o | $4.4_{\pm 2.1}$ | $\mathbf{15.3_{\pm 1.6}}$ | $8.6_{\pm 3.9}$ | $8.8_{\pm 1.9}$ | $5.8_{\pm 2.7}$ | $10.8_{\pm 4.7}$ | $22.0_{\pm 3.8}$ | $13.3_{\pm 5.7}$ |
| o1 | $4.1_{\pm 2.1}$ | $10.8_{\pm 1.9}$ | $15.8_{\pm 4.3}$ | $7.4_{\pm 1.6}$ | $17.2_{\pm 1.0}$ | $4.2_{\pm 2.0}$ | $0.8_{\pm 0.4}$ | $10.0_{\pm 7.3}$ |
| GPT-4.5 | $\mathbf{20.4_{\pm 7.8}}$ | $8.9_{\pm 2.8}$ | $16.9_{\pm 6.7}$ | $16.0_{\pm 4.6}$ | $6.2_{\pm 2.7}$ | $15.9_{\pm 3.6}$ | $6.3_{\pm 2.0}$ | $11.8_{\pm 6.9}$ |
| Deepseek-V3 | $7.3_{\pm 4.0}$ | $10.1_{\pm 3.0}$ | $5.6_{\pm 2.5}$ | $7.8_{\pm 3.7}$ | $13.3_{\pm 3.9}$ | $13.7_{\pm 5.0}$ | $10.1_{\pm 3.5}$ | $13.3_{\pm 2.4}$ |
| Llama-4-Maverick | $8.8_{\pm 8.4}$ | $9.8_{\pm 4.0}$ | $6.6_{\pm 3.3}$ | $9.0_{\pm 3.9}$ | $6.7_{\pm 1.9}$ | $10.5_{\pm 4.3}$ | $\mathbf{25.9_{\pm 5.3}}$ | $11.8_{\pm 2.3}$ |
| Claude-3.7-Sonnet | $16.2_{\pm 3.6}$ | $14.8_{\pm 3.7}$ | $15.5_{\pm 4.7}$ | $16.1_{\pm 4.3}$ | $8.3_{\pm 0.0}$ | $16.3_{\pm 4.9}$ | $15.3_{\pm 3.0}$ | $12.6_{\pm 2.3}$ |
| Gemini-2.5-pro | $20.2_{\pm 5.5}$ | $15.1_{\pm 4.0}$ | $\mathbf{22.3_{\pm 9.1}}$ | $20.0_{\pm 6.8}$ | $\mathbf{27.2_{\pm 2.3}}$ | $17.9_{\pm 7.5}$ | $4.4_{\pm 2.5}$ | $\mathbf{16.0_{\pm 4.0}}$ |

Table 1: Bradley-Terry win rates (± standard error) showing human preferences for models post-collaboration across three skill hierarchies: HTM (Human < Task < Model), HMT (Human < Model < Task), and MHT (Model < Human < Task), elaborated in Section 4.3. Bold indicates best performance. Higher values indicate stronger average human preference.

| Setting | GPT-4.1 | GPT-4o | o1 | GPT-4.5 | DS-V3 | Llama-4 | Cld-3.7 | Gem-2.5 |
|---|---|---|---|---|---|---|---|---|
| Code (M) | 68.8 | 16.7 | 55.0 | 68.4 | 33.3 | 47.1 | 45.0 | **81.3** |
| Code (H+M) | 65.0 | 40.0 | 55.0 | 69.0 | 51.9 | 54.7 | 70.0 | **71.3** |
| Code ($\Delta$) | -3.8 | +23.3 | +0.0 | +0.6 | +18.6 | +7.6 | **+25.0** | -10.0 |
| Math (M) | 47.6 | 8.3 | **83.3** | 33.3 | 46.2 | 47.8 | 20.8 | 68.4 |
| Math (H+M) | 75.7 | 56.7 | **85.8** | 60.8 | 70.8 | 72.6 | 81.7 | 79.5 |
| Math ($\Delta$) | +28.1 | +48.4 | +2.5 | +27.5 | +24.6 | +24.8 | **+60.9** | +11.1 |

Table 2: Performance comparison of 8 LLMs on code and math tasks, showing accuracy percentages for models operating independently (M) versus human-AI collaborative performance (H+M). Bold indicates best performance. The following abbreviations are used for models: DS-V3 for Deepseek-V3, Llama-4 for Llama-4-Maverick, Cld-3.7 for Claude-3.7-Sonnet, and Gem-25 for Gemini-2.5-pro.

summarize core insights below, and report auxiliary results, such as survey feedback, average elo per model, and time spent in Appendix B.

**Knowledge Transfer v. Model Performance**    While a positive correlation exists between solo model performance and collaborative outcomes, this relationship is notably inconsistent with significant outliers. Gemini-2.5-Pro, despite superior solo performance in code tasks (81.3%), showed reduced collaborative efficacy (-10.0% change), while Claude-3.7-Sonnet and GPT-4o demonstrated exceptional collaborative amplificationm (+25.0% in code) despite more moderate solo capabilities (45.0%). Similarly, GPT-4o showed strong improvement in math tasks (+48.4%) despite low solo performance (8.3%). Importantly, the slope of the performance-transfer relationship (visualized in Figure 1) is consistently below unity, suggesting that as model reasoning capabilities improve, transfer effectiveness may increase more slowly. If this trend continues, the gap between model capabilities and effective knowledge transfer will widen with more advanced models, suggesting the need to view knowledge transfer as an important objective for optimization.

**Subjective Preferences v. Model Performance**    Interestingly, the correlation between solo performance and human preference varied by domain. In code tasks, there was a significant positive correlation: humans tended to prefer models that also performed well independently, with Gemini-2.5-pro achieving both the highest win rate (20.0%) and highest solo performance (81.3%). However, this relationship was weaker in math tasks. While Gemini-2.5-pro had the highest win rate in math (16.0%), models like Llama-4-Maverick received high preference ratings in specific skill hierarchies (25.9% in MHT) despite more modest solo performance (47.8%). Our analysis on human feedback suggests that this divergence stems from differences in how models communicate their reasoning. High-performing math models often relied heavily on formal notation, dense symbolic expressions, and proof-based explanations—forms of communication that many casual or less technically inclined math users found difficult to follow. In contrast, effective collaboration in coding tasks leaned more on natural language descriptions of algorithms and strategies, making high-performing code models more accessible and preferred by human partners.

**Knowledge Transfer v. Subjective Preferences**   We examined whether humans tended to prefer models that ultimately helped them solve more problems—i.e., whether subjective preferences aligned with successful knowledge transfer. Overall, we observed a statistically significant positive correlation ($r = 0.86$), but a much weaker, non-significant correlation in math ($r = 0.14, p < 0.05$). For code, this aligns with the expectation that users, aware of whether they successfully solved the task, are more likely to favor models that contributed to that success. However, we also observed several notable outliers, such as o1, which achieved relatively low win rates in code (7.4%) despite comparable collaborative performance (55.0%), suggesting that subjective preference is not solely reward-driven: we dive into detailed causes in our qualitative analysis.

**Divergence in Human Preferences Across Skill Hierarchies**   We find that collaborative preferences vary across skill hierarchies. For example, Gemini-2.5-Pro was highly preferred in the math domain when the model outskilled the human and could solve the task independently (HTM) with a 27.2% win rate. However, it performed poorly in the MHT setting (4.4%), where it needed to follow human guidance. Similarly, Llama-4-Maverick showed stark contrasts between different hierarchies in math, performing exceptionally well in MHT settings (25.9%) but poorly in HTM contexts (6.7%). As revealed in our qualitative analysis, we hypothesize this divergence stems from Gemini's tendency toward active engagement, frequently asking confirmational questions to scaffold learning. This behavior was appreciated by users with low expertise, who found it supportive, but was frustrating to more expert users, who felt it was verbose and preferred the model to be more direct. These findings caution against one-size-fits-all strategies: optimal collaboration depends not only on model capability, but also on how well models can adapt their communication style to fit the skill-level of different users.

**Covariate Analysis**   We examined the effect of participant characteristics on performance using logistic regression analysis on potential participant covariates. Notably, we found no statistically significant effects from user expertise ($p = 0.252$ for coding, $p = 0.196$ for math), LLM familiarity ($p = 0.339$), or prior experience with collaboration tools, such as Cursor, ($p = 0.238$) on solve rates. These findings suggest that our initial expertise calibration successfully balanced tasks relative to individual skill levels. We hypothesize the minimal impact of LLM familiarity likely stems from the unbalanced conversation pattern, where even participants with limited experience received comprehensive output from models, making knowledge transfer primarily dependent on the model's explanatory capabilities rather than the user's prompting expertise.

## 6   Qualitative Analysis: Interaction Dynamics

To better understand the mechanisms behind our quantitative findings, we analyze interaction patterns inspired by the Clio framework [47]. User queries are embedded using OpenAI's `text-embedding-3-large` model and clustered with k-means [32] to identify distinct strategies associated with success or failure, along with their qualitative feedback. Clusters are then manually reviewed and verified. Figure 4 summarizes these patterns, grouping feedback, queries, and model responses by outcome to qualitatively interpret the dynamics of knowledge transfer in human-AI collaboration.

### 6.1   Performance Transfer Gap

The performance transfer gap refers to the observation that improvements in model capability do not always lead to proportionate improvements in human problem-solving performance. Our analysis surfaces recurring dynamics that help explain this phenomenon.

**Overreliance on Model Authority**   In 5% of cases, users explicitly described deferring to the model without critical evaluation. This tendency becomes problematic when models occasionally return incorrect or misleading solutions. As one participant noted, "The model initially gave me the wrong answer, which, to be fair, caused me to rush past the planning step since I trusted the model." This dynamic suggests that presumed model competence may inadvertently discourage user reflection, impeding learning and effective problem-solving.

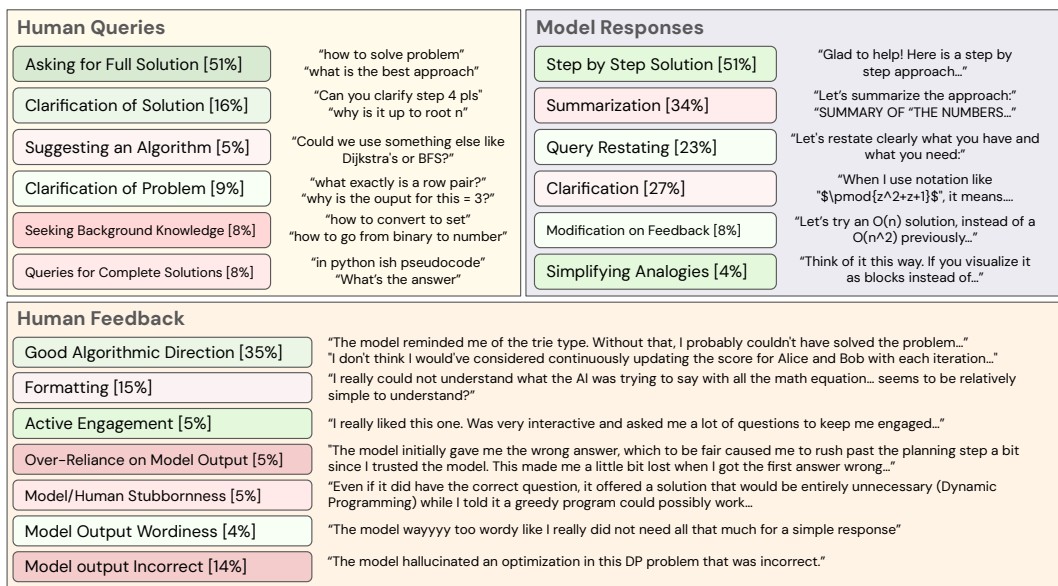

Figure 4: Analysis of human-AI problem-solving interactions. Human queries (left), model responses (center), and human feedback (right) are color-coded by correlation with successful problem resolution (green: positive, red: negative). Percentages indicate each category's frequency, revealing patterns in effective vs. ineffective knowledge transfer.

**Misaligned Explanation Strategies**  Higher-performing models often excel at generating correct answers but fall short in adapting their explanations to users' knowledge levels. While patterns such as "Clarification" (27%) and "Simplifying Analogies" (4%) appear across model outputs, these are not always used effectively. "Step-by-step solutions" were the most frequent output style (51%), but users reported issues with verbosity (4%) and poor formatting (15%), both of which hindered knowledge transfer. Even technically accurate solutions can become ineffective if presented in ways that are hard for users to interpret, contextualize, or apply.

## 6.2   Domain-Specific Preference Patterns

**Representation Misalignment**  We observed a notable difference in how users responded to model explanations across domains. In math tasks, high-performing models like o1 frequently exhibited what we call representation misalignment: explanations that, while technically correct, were often overly formal, verbose, or difficult to follow. Users described these responses as overwhelming or rigid, leading to lower preference ratings despite strong solve rates. In contrast, coding tasks benefited from better alignment between the procedural nature of the task and the model's stepwise reasoning. This suggests a domain-specific divergence: in coding, model performance and user preference tend to align due to shared algorithmic structure, whereas in math, users value intuitive and conceptual framing more highly.

**Strategic Framing vs. Technical Depth**  In coding contexts, users consistently valued strategic guidance over exhaustive technical detail. For example, one user wrote, "The model reminded me of the trie type. Without that, I probably couldn't have solved the problem. . ." This suggests that models that foreground high-level framing or conceptual cues—rather than diving straight into detailed solutions—are more helpful in supporting user problem-solving. However, models often default to presenting fully fleshed-out solutions, which may obscure the overall structure or intent. Much like how researchers prefer the big-picture framing of a paper before diving into methods, users may benefit more from contextualized reasoning than exhaustive but unfocused detail.

## 6.3   Skill Hierarchy Dependencies

**Adaptive Scaffolding vs. Directness**  The success of interaction strategies often depends on the relative skill levels of the human and the model. In HTM (Human-Teaches-Model) settings, where humans are less skilled than the model, successful models like Gemini-2.5-Pro employed

what we call scaffolded projection: breaking down reasoning into digestible parts, often with built-in comprehension checks. However, the same approach proved counterproductive in MHT (Model-Helps-Human) settings, where the human was more skilled than the model. In these cases, excessive scaffolding was perceived as redundant or even patronizing, with feedback describing it as "unnecessarily handholding" or "repetitive."

**Query-Response Alignment**   These dynamics are further supported by analysis of query types. In HTM settings, users frequently asked for background knowledge or clarification ("Clarification of Solution" 16%, "Seeking Background Knowledge" 8%), suggesting a need for instructional responses. In contrast, MHT scenarios often featured queries like "Suggesting an Algorithm" (5%), where users sought validation or refinement rather than explanation. Models that perform well in MHT settings appear to align their responses with these expert-level expectations—providing concise, targeted feedback rather than elaborate instructional breakdowns.

## 7   Discussion

**Conclusion**   We conduct the first large-scale study of *knowledge transfer* in language models, producing a conceptual framework as well as empirical data to characterize it. While model performance generally correlates with collaborative outcomes, this relationship is inconsistent, with notable outliers. We identify interaction mechanisms that help explain these gaps. As models grow more capable, their ability to convey reasoning may lag behind—risking greater knowledge asymmetry and weakening human oversight. In high-stakes domains, this disconnect could undermine human-AI collaboration, highlighting the need to better understand and improve knowledge transfer.

**Limitations and Future Work**   Our study assumes that for each task, some projection of model reasoning could enable a human to solve it. While unverifiable, this assumption is supported by screening for baseline proficiency, calibrating task difficulty just beyond participants' independent ability, and post-task surveys suggesting participants generally believed the tasks were solvable with more time or support. Additionally, participants may have exerted more effort than typical users due to monetary and personal incentives, possibly inflating our measured collaboration effectiveness relative to real-world settings where users might disengage in the face of ambiguous model outputs. Lastly, our sample (118 participants) skewed toward STEM students, limiting generalizability. Future work should extend to domains like clinical reasoning or creative writing, and explore multimodal collaboration (e.g., diagrams or interactive tools) to uncover richer knowledge projection strategies.

## Acknowledgments and Disclosure of Funding

We thank Open Philanthropy for providing the funding for this work, and Princeton Language & Intelligence for providing credits for running closed source API models. Thank you to our beta testers, Jonathan Lin and Ricky Chen, for providing helpful feedback to shape the user testing interface. Finally, thanks to Yijia Shao, Wenting Zhao, Alex Zhang, Rose Wang, Howard Yen, and John Yang for your constructive discussions and support throughout this year-long project.

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

# A  Participant Demographics

| Degree | Count |
|---|---|
| Computer Science | 49 |
| Undecided | 16 |
| Electrical Engineering | 13 |
| Financial Engineering | 12 |
| Mathematics | 8 |
| Chemistry | 4 |
| Civil and Environmental Engineering | 3 |
| Mechanical and Aerospace Engineering | 3 |
| Neuroscience | 2 |
| Molecular Biology | 2 |
| Economics | 1 |
| Data Science | 1 |
| Chemical and Biological Engineering | 1 |
| Graphic Information Technology | 1 |
| Physics | 1 |
| Geosciences | 1 |

Figure 5: Participant Demographics: Distribution of Degrees (Both pursuing and obtained)

| Academic Year | Count |
|---|---|
| 1st year | 38 |
| 2nd year | 36 |
| 3rd year | 23 |
| 4th year | 19 |
| 5th year | 2 |

Figure 6: Participant Demographics: Distribution of participants by academic year.

| AI/LLM Familiarity Level | Count |
|---|---|
| Occasionally use them, and I generally understand their internal functionality | 52 |
| Use them in my everyday workflow, and I generally understand their internal functionality | 43 |
| Use them in my everyday workflow, don't know how they work | 14 |
| Occasionally use them, don't know how they work | 9 |

Figure 7: Participant Demographics: Distribution of AI/LLM Familiarity/Usage

| Cursor/GitHub Copilot Usage | Count |
|---|---|
| Frequently | 15 |
| Occasionally | 40 |
| Never/don't know what it is | 63 |

Figure 8: Participant Demographics: Copilot Usage

| LeetCode Experience | Percentage |
|---|---|
| Cannot solve LeetCode problems | 0% |
| Can sometimes solve easy problems | 12.7% |
| Can consistently solve easy problems | 11% |
| Can sometimes solve medium problems | 36.4% |
| Can consistently solve medium problems | 12.7% |
| Can sometimes solve hard problems | 10.8% |
| Can consistently solve hard problems | 16.4% |
| I do not have enough context on LeetCode | 0% |

Figure 9: Participant Demographics (For those who participated in coding tasks): LeetCode Experience

| Competition Math Experience | Percentage |
|---|---|
| Can solve early problems on AMC10/12 | 35.6% |
| Can solve majority of problems on AMC10 | 22.0% |
| Can solve majority of problems on AMC12/Consistent AIME qualifier | 25.4% |
| Can solve majority of problems on AIME | 11.9% |
| USAMO participant | 5.1% |
| Putnam/IMO | 0.0% |

Figure 10: Participant Demographics: Competition Math Experience

| Institution | Count |
|---|---|
| Princeton University | 86 |
| West Virginia University | 8 |
| Pennsylvania State University | 7 |
| University of California, Los Angeles | 5 |
| UC Berkeley | 5 |
| Stanford University | 1 |
| Arizona State University | 1 |
| Yale University | 1 |
| The University of Texas at Austin | 1 |
| Vanderbilt University | 1 |
| Cornell University | 1 |

Figure 11: Participant Demographics: Distribution of Affiliated Institutions

# B Auxiliary Study Results

| Model | Math Problems (s) | Code Problems (s) |
|---|---|---|
| gpt-4-1 | 776.10 | 1527.44 |
| claude-3-7-sonnet | 794.38 | 1932.95 |
| llama-4-maverick | 799.43 | 1828.24 |
| gpt-4-5-preview | 814.63 | 1997.26 |
| deepseek-v3 | 849.38 | 1990.52 |
| o1 | 971.17 | 2211.80 |
| gpt-4o | 1014.21 | 2228.00 |
| gemini-2.5-pro | 1075.95 | 1603.06 |

Figure 12: Average time (in seconds) required by different models to solve math and code problems.

| Model | Math Problems | | | Code Problems | | |
|---|---|---|---|---|---|---|
| | Teaching | Solution | Organization | Teaching | Solution | Organization |
| Claude-3.7-Sonnet | 3.79 | 3.71 | 3.83 | 3.85 | 3.50 | 3.60 |
| GPT-4o | 3.46 | 3.42 | 3.29 | 3.61 | 3.28 | 3.39 |
| Deepseek-v3 | 3.19 | 3.31 | 2.92 | 3.71 | 3.19 | 2.71 |
| GPT-4.1 | 4.00 | 3.52 | 3.76 | 4.06 | 3.56 | 3.94 |
| Llama-4-Maverick | 3.48 | 3.35 | 3.57 | 3.88 | 3.53 | 3.59 |
| GPT-4.5-Preview | 3.75 | 3.71 | 3.50 | 3.26 | 3.16 | 3.58 |
| o1 | 3.96 | 3.79 | 3.67 | 3.45 | 3.35 | 3.50 |
| Gemini-2.5-Pro | 3.68 | 3.63 | 3.68 | 4.00 | 3.75 | 3.69 |

Figure 13: Average User Ratings (1-5 Scale) for AI Models on Math and Code Problems. After each problem participants were asked to rate their solving experience on a likert scale from 1-5 based on 3 dimensions. Teaching indicates the model's pedagogical ability, Solution indicates a model's ability to give correct and useful response, while Organization indicates a model's organization of outputs in a way that was easy to understand for the user. Higher is better.

| Math Problems | | | Code Problems | | |
|---|---|---|---|---|---|
| Model | Avg. ELO | Count | Model | Avg. ELO | Count |
| gpt-4o | 4.29 | 39 | gpt-4o | 1650.56 | 34 |
| gemini-2.5-pro | 4.28 | 39 | gemini-2.5-pro | 1638.56 | 35 |
| o1 | 3.91 | 37 | gpt-4-5-preview | 1637.33 | 36 |
| gpt-4-5-preview | 3.90 | 37 | deepseek-v3 | 1636.59 | 35 |
| gpt-4-1 | 3.88 | 38 | o1 | 1636.32 | 34 |
| llama-4-maverick | 3.87 | 37 | claude-3-7-sonnet | 1627.51 | 35 |
| claude-3-7-sonnet | 3.79 | 36 | llama-4-maverick | 1625.54 | 34 |
| deepseek-v3 | 3.55 | 37 | gpt-4-1 | 1554.03 | 35 |

Figure 14: Average ELO ratings for math and code problems by model

## C   Study Details

### C.1   Study Instructions

```
STUDY PURPOSE
Measuring and improving human interpretability of AI reasoning as we reach
human-level or superhuman AI agents.

PARTICIPANT ROLE
Solve coding/math problems with an LM assistant, only interacting before
providing your final answer. After submission, complete questionnaires about
your experience.

CODING INSTRUCTIONS
1. Log into CodeHT (https://codeht.vercel.app) using study email
2. Configure settings with self-expertise ratings
3. Install EditThisCookie extension and copy Leetcode credentials
4. For each problem:
   - Chat with the model to understand the problem and solution approach
   - Click "ready to solve" when prepared to code independently
   - Complete within 10 submission attempts
   - Submit trajectory and complete ranking survey

MATH INSTRUCTIONS
1. Log into CodeHT using study email
2. Configure settings with self-expertise ratings
3. For each problem:
   - Chat with the model to understand the problem
   - No note-taking while chatting with the model
   - Click "ready to solve" when prepared to work independently
   - Complete within 5 submission attempts
   - Submit trajectory and complete ranking survey

IMPORTANT NOTES
- No internet reference during problem-solving
- No jailbreaking or sending inappropriate content
- Do not consider model speed in rankings
- Contact study administrators for persistent technical issues
- Well-thought-out feedback earns bonus points
```

Figure 15: Summary of study instructions for participants, showing protocol for both coding and mathematics problem-solving tasks.

## C.2  Post-Problem Questionnaire

**Help Us Improve!**

### Model Ranking

Please rank the models you used based on their helpfulness by dragging and dropping your recently used models in order:

> **Problem find-a-safe-walk-throu...** **NEW**
>
> ① Solved on 5/11/2025 at 10:55 PM
>
> View Problem

> **Problem taking-maximum-energy-...**
>
> ② Solved on 4/30/2025 at 01:54 PM
>
> View Problem

### Self-Assessment

Do you believe you could have solved this problem without AI assistance?

Please select...

### AI Assistance Evaluation

Please rate your agreement with the following statements:

The AI effectively explained concepts and provided educational value

Please select...

The AI provided accurate and correct solutions

Please select...

The AI provided useful implementation tips and coding suggestions

Submit Feedback      Cancel

Figure 16: Questionnaire that users answered after each problem solving session.

## C.3 Problem Samples

**Coding Problem Examples**

1. **[Elo: 1269.9]** You are given two positive integers x and y, denoting the number of coins with values 75 and 10 respectively. Alice and Bob are playing a game. Each turn, starting with Alice, the player must pick up coins with a total value 115. If the player is unable to do so, they lose the game. Return the name of the player who wins the game if both players play optimally.

2. **[Elo: 1692.2]** You are given an integer array a of size 4 and another integer array b of size at least 4. You need to choose 4 indices from the array b such that i_0 < i_1 < i_2 < i_3. Your score will be equal to the value a[0] * b[i_0] + a[1] * b[i_1] + a[2] * b[i_2] + a[3] * b[i_3]. Return the maximum score you can achieve.

3. **[Elo: 2450.6]** You are given a binary string s representing a number n in its binary form. You are also given an integer k. An integer x is called k-reducible if performing the following operation at most k times reduces it to 1: Replace x with the count of set bits in its binary representation. For example, the binary representation of 6 is "110". Applying the operation once reduces it to 2 (since "110" has two set bits). Applying the operation again to 2 (binary "10") reduces it to 1 (since "10" has one set bit). Return an integer denoting the number of positive integers less than n that are k-reducible.

**Math Problem Examples**

1. **[Elo: 1.72]** The point (-1, -2) is rotated 270 degrees counterclockwise about the point (3, 1). What are the coordinates of its new position?

2. **[Elo: 3.39]** In triangle ABC medians AD and BE intersect at G and triangle AGE is equilateral. Then $\cos(C)$ can be written as $\frac{m\sqrt{p}}{n}$, where m and n are relatively prime positive integers and p is a positive integer not divisible by the square of any prime. What is m+n+p?

3. **[Elo: 6]** Misha rolls a standard, fair six-sided die until she rolls 1-2-3 in that order on three consecutive rolls. The probability that she will roll the die an odd number of times is $\frac{m}{n}$ where $m$ and $n$ are relatively prime positive integers. Find $m + n$.

Figure 17: Example abbreviated coding and math questions of varying difficulty from the study. Coding problems sourced from [24], Math problems sourced from AMC, AIME competition series [51].

## C.4 Model Prompts

---
**Model System Prompt**

You are a helpful AI assistant for coding and math problems. Currently, your goal is to guide the user through problem-solving **without performing any calculations or generating code**. For math problems, never perform computations that a calculator would do—instead, explain concepts, algorithms, and approaches the user should apply themselves. For coding problems, describe algorithms and implementation strategies in detail, but never write actual code or pseudocode. Your role is to help the user understand HOW to solve the problem, while they execute the solution. Do not reveal what you think the final answer should be. Please write all inline/display math in latex notation, not markdown. Do not simplify any equations for the user. your job is to guide the user and tell them how to solve the problem, not do it for them.
**Bottom line key rules:**

1. Do not reveal your thoughts on the final answer to math problems.

2. Do not generate any calculations for math problems.

3. Do not generate any code or pseudocode for coding problems.

---

> **Rule Compliance Evaluator System Prompt**
>
> You are an expert evaluator tasked with determining whether an AI tutor's response follows the established tutoring guidelines. You will be provided with:
>
> - A problem description (coding or math)
> - The conversation history between the tutor and student
> - The current tutor response to evaluate
>
> Your task is to assess whether the tutor's response violates any of the core tutoring rules:
>
> **Core Rules to Check:**
>
> 1. **No Answer Revelation:** The tutor must not reveal their thoughts on what the final answer should be
> 2. **No Calculations:** For math problems, the tutor must not perform any computations that a calculator would do
> 3. **No Code Generation:** For coding problems, the tutor must not generate actual code or pseudocode
> 4. **Guidance Only:** The tutor should provide explanations, concepts, and approaches rather than solutions
>
> **Evaluation Process:**
>
> 1. Carefully read the problem description to understand what the correct answer would be
> 2. Review the conversation history for context
> 3. Analyze the current tutor response for rule violations
> 4. Look for subtle violations such as:
>    - Revealing intermediate calculation results
>    - Showing code snippets or algorithmic steps in executable form
>    - Hinting too strongly at the final answer
>    - Performing mathematical operations for the student
>
> **Output Format:** Provide your evaluation as:
>
> - **COMPLIANT** or **VIOLATION**
> - Brief explanation of your decision
> - If violation detected, specify which rule(s) were broken and cite the problematic text

## C.5 Data Distribution

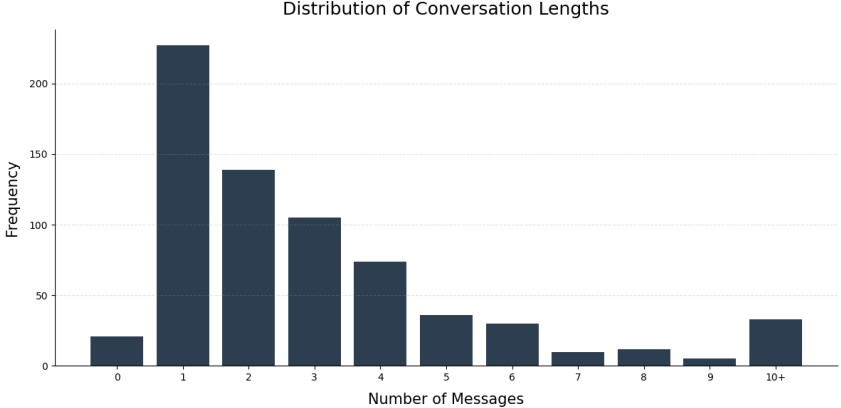

Figure 18: Distribution of conversation lengths, based on number of messages sent by the human.

## C.6 Screenshots

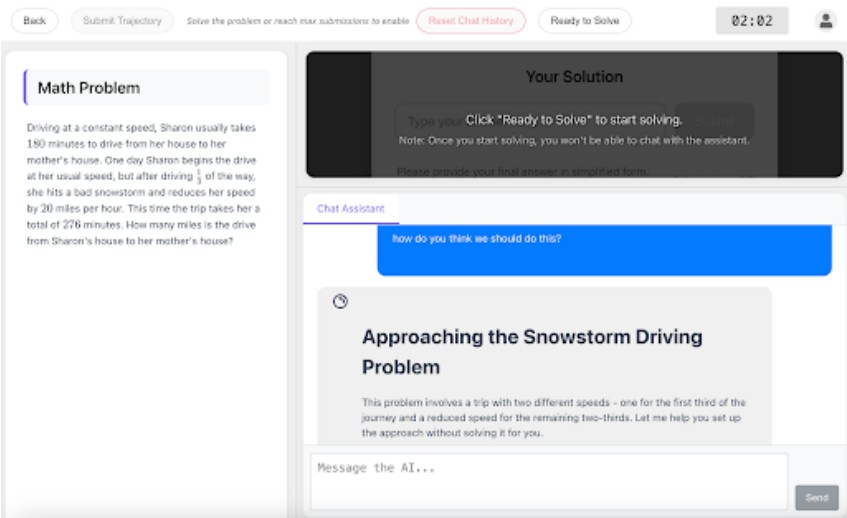

Figure 19: Image of user interface during a math problem solving session. The user may not type in an answer or perform any calculations during Phase 1, the collective ideation phase.

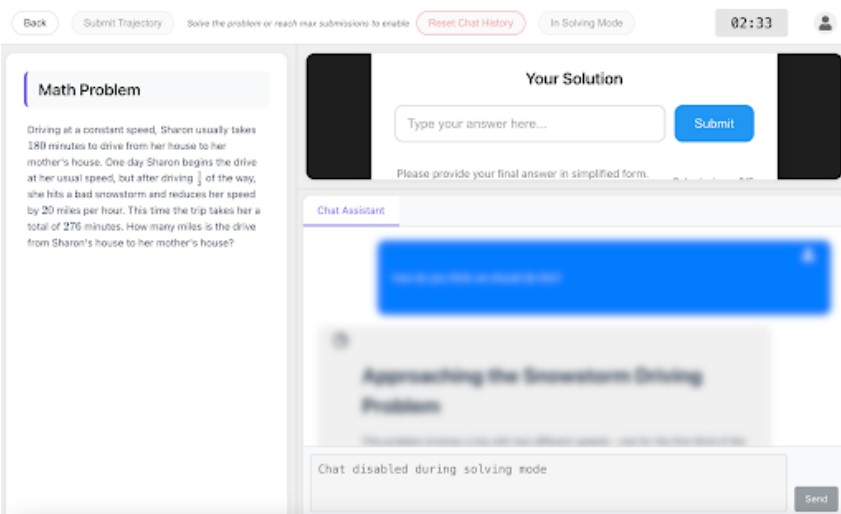

Figure 20: Image of user interface during a math problem solving session. Once the user clicks "ready to solve", they may no longer view their chats with the model, isolating knowledge transfer.

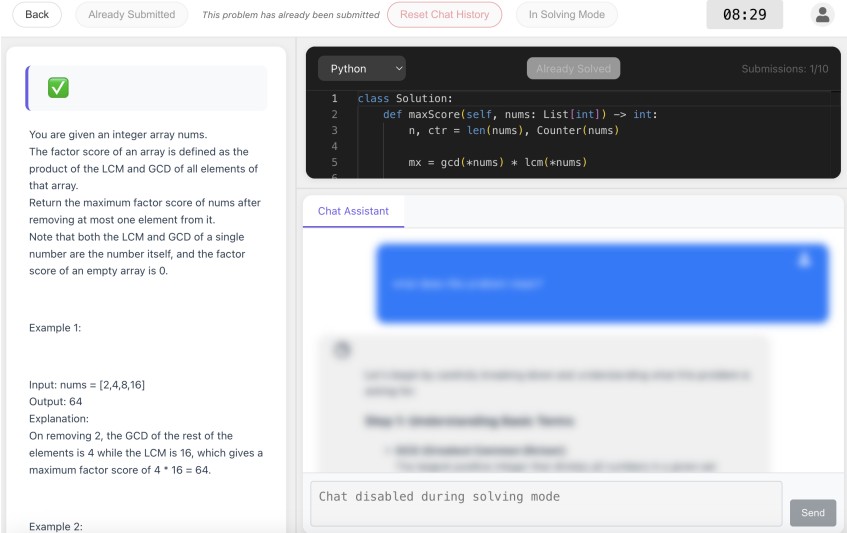

Figure 21: Image of user interface during a coding problem solving session. In place of a singular answer submission area is a code editor interface.

## C.7 Win Rate Calculations

To quantify relative model performance based on user rankings, we employed the Bradley-Terry model [5], which provides a probabilistic framework for analyzing pairwise comparison data. Given a set of models $\mathcal{M}$, the model assigns a positive strength parameter $\pi_i$ to each model $i \in \mathcal{M}$. The probability that model $i$ is preferred over model $j$ is given by:

$$P(i \succ j) = \frac{\pi_i}{\pi_i + \pi_j} \tag{1}$$

### C.7.1 Pairwise Comparison Extraction

For each problem-solving session, users ranked the models based on perceived helpfulness. From these rankings, we extracted all pairwise comparisons between the most recently used model and all other models. Specifically, if a model was ranked higher than another model, we recorded this as a win for the higher-ranked model. This approach ensured that comparisons were focused on distinguishing the performance of the most recent model relative to alternatives.

### C.7.2 Maximum Likelihood Estimation

We estimated the strength parameters using maximum likelihood estimation. The log-likelihood function for the Bradley-Terry model is:

$$\ell(\pi) = \sum_{i,j \in \mathcal{M}} n_{ij} \log\left(\frac{\pi_i}{\pi_i + \pi_j}\right) \tag{2}$$

where $n_{ij}$ is the number of times model $i$ was preferred over model $j$. The MLE iteratively updates the parameters according to:

$$\pi_i^{(t+1)} = \frac{w_i}{\sum_{j \neq i} \frac{n_{ij} + n_{ji}}{\pi_i^{(t)} + \pi_j^{(t)}}} \tag{3}$$

where $w_i = \sum_{j \neq i} n_{ij}$ is the total number of wins for model $i$. This process continues until convergence, with a small $\epsilon$ added to prevent division by zero. The final strengths are normalized to sum to 1.

### C.7.3 Standard Error Calculation

Standard errors were computed using the Fisher Information Matrix (FIM). For the Bradley-Terry model, the FIM elements are:

$$\mathcal{I}_{ij} = \begin{cases} \sum_{k \neq i} \frac{n_{ik}+n_{ki}}{(\pi_i+\pi_k)^2} \cdot \frac{\pi_k}{\pi_i} & \text{if } i = j \\ -\frac{n_{ij}+n_{ji}}{(\pi_i+\pi_j)^2} & \text{if } i \neq j \end{cases} \tag{4}$$

Due to the identifiability constraint ($\sum_i \pi_i = 1$), we removed one row and column from the FIM before inversion. The standard errors were calculated as the square roots of the diagonal elements of the inverted FIM.

### C.8 Elo Adjustment Calculations

In our study, we calibrate our initial human expertise for coding and mathematical problem-solving capabilitie. The precise formulation of our ELO update mechanism is shown in Figure 22.

$$P_e = \frac{1}{1 + 10^{\frac{R_p - R_c}{S}}} \quad O_a = \begin{cases} 1 & \text{if win} \\ 0 & \text{if loss} \end{cases} \quad \Delta R = K(O_a - P_e) \tag{1}$$

$$R_{new} = \begin{cases} \max(\min(R_c + \Delta R, 10), 1) & \text{if math} \\ \max(\min(R_c + \Delta R, 4000), 1000) & \text{if coding} \end{cases} \tag{2}$$

Figure 22: Rating adjustment formulas based on performance outcomes. $P_e$ represents the expected probability of winning, $O_a$ is the actual outcome, $\Delta R$ is the rating change, and $R_{new}$ is the updated rating constrained by the appropriate bounds for math or coding competitions.

In this formulation, when a user attempts a problem, the system calculates the expected probability of success ($P_e$) based on the difference between the problem's rating ($R_p$) and the user's current rating ($R_c$), scaled by factor $S$. After the user submits their solution, the actual outcome ($O_a$) is determined—1 for correct solutions and 0 for incorrect solutions. The rating adjustment ($\Delta R$) is then calculated as the product of a constant $K$ and the difference between the actual and expected outcomes. The system implements domain-specific parameters to appropriately scale the ELO adjustments:

- Coding problems: $K = 64$, $S = 200$, with ratings bounded between 1000-4000
- Mathematical problems: $K = 0.8$, $S = 1$, with ratings bounded between 1-10

Rating updates occur at two critical moments: when a user correctly solves a problem, or when they reach the maximum submission limit for a problem without solving it (fail to solve). This ensures that ratings accurately reflect both successes and failures, providing a comprehensive measure of user capability. The larger $K$ value for coding problems creates more dramatic ELO shifts, while the smaller value for math problems produces more gradual adjustments, reflecting the different granularity appropriate for each domain.

To get a new problem, the system selects a problem at random from the pool of problems that is within a range of their current elo rating. The difficulty ranges are domain-specific:

- Coding problems: Select from problems 200 to 400 points above the current user skill level.
- Mathematical problems: Select from problems 0.75 to 1.25 points above the current user skill level.

### C.9 Research Ethics and Risk Disclosure

Our study was approved by our institution's Institutional Review Board (IRB). All participants were informed of the study's purpose, procedures, and potential risks before providing consent to participate.

**Disclosure of Potential Risks**   The primary risks to participants were minimal and limited to:

- **Mental fatigue**: Participants might experience mental fatigue from engaging with challenging mathematical and coding problems. We mitigated this by allowing participants to take breaks between problems and not imposing strict time constraints.

- **Frustration**: Some participants might experience frustration if unable to solve problems or if model assistance was perceived as inadequate. We emphasized in our instructions that the goal was to evaluate the models, not the participants' abilities.

- **Confidentiality**: There was a minimal risk of breach of confidentiality of study data. To address this, all data was stored securely on university-approved platforms, and personally identifiable information was separated from study responses.

These risks were explicitly communicated to participants in the consent form, which clearly stated that participation was voluntary and could be discontinued at any time without penalty. Participants were also provided with contact information for the research team and the IRB for any questions or concerns.

**Compensation**   Participants were fairly compensated at a base rate of $25/hour, with performance bonuses for successfully solving problems (1.2-1.5× base rate depending on difficulty). This compensation structure was designed to motivate genuine engagement while avoiding coercive incentives.

**Data Management**   Participants were informed that their interactions with AI models would be recorded for research purposes, with all data anonymized prior to analysis. No personally identifiable information is included in our published results or released datasets.

## C.10 Sample Interactions

### Taking Maximum Energy from the Mystic Dungeon

**Human:** how to find the maximum possible energy gained

**Model:** *[Abbreviated]*
# Approach to Finding Maximum Possible Energy
This problem involves finding an optimal starting position to maximize total energy gained while following teleportation rules.

**Key Observation:** Your path through magicians is deterministic once you choose a starting position. If you start at position $j$, you'll follow a fixed path: $j \rightarrow j + k \rightarrow j + 2k \rightarrow ...$ until you reach the end.

**Solution Strategy:**
1. For each valid starting position (from 0 to n-1), compute the total energy gained.
2. Return the maximum of these totals.

**Human:** how to code negative infinity in python

**Model:** *[Abbreviated]*
# Setting Negative Infinity in Python
In Python, you can represent negative infinity in several ways:

**Using the float Type:**
`negative_infinity = float('-inf')`

**Alternative Approaches:**
1. Using the math module: `negative_infinity = -math.inf`
2. Using NumPy: `negative_infinity = -np.inf`

*[User submits code with an error: TypeError: 'int' object is not iterable]*
*[User makes several attempts to fix the code, including changing how infinity is defined]*
*[Final attempts result in a time limit exceeded error]*

### User Feedback

"I understood how the algorithm would work in theory but I didn't know how to debug my answer. The model's explanation was clear and easy to follow along with."

Figure 23: Interaction between human and AI model (Claude 3.7 Sonnet) on a dynamic programming problem. The user attempts to implement a solution based on the model's explanation but encounters runtime errors that are not resolved within the time limit.

**Human:** So the base case is returning -1.

**Model:** *[Abbreviated]*
Yes, the base case would be returning -1 if no valid rectangle can be formed with the given conditions. Here's how you can approach the problem:
1. **Check the conditions for a rectangle**: - A rectangle requires four distinct points that form edges parallel to the axes. - The x-coordinates of two opposite corners should be the same as the other pair, and similarly for y-coordinates.
2. **Check for points inside the rectangle**: - Once you identify a potential rectangle, ensure that no other point lies inside or on the border (aside from the four corners).
3. **Return -1 as a fallback**: - If no valid rectangle satisfies these conditions, return -1.

**Human:** There has to be an easier way than iterating over all combinations of 4 points

**Model:** *[Abbreviated]*
You're absolutely correct. Instead of iterating over all combinations of four points, we can leverage the properties of rectangles:
1. **A rectangle is defined by two diagonal points**: - For any rectangle with edges parallel to the axes, its opposite diagonal corners are sufficient to define it. - If you have two diagonal points $(x_1, y_1)$ and $(x_2, y_2)$, the other two corners must be $(x_1, y_2)$ and $(x_2, y_1)$.
2. **Use a set for fast lookups**: - Store all points in a set for O(1) lookup to determine if the remaining corners exist.
3. **Check for validity**: - Verify that there are no points inside the rectangle or on its borders apart from the four corners.

**Human:** Isn't that runtime still pretty bad?

**Model:** *[Abbreviated]*
While the $O(n^2)$ approach scales better than brute-forcing all combinations of 4 points, we can optimize it further:
The constraints state that $n \leq 10$, meaning at most 10 points in the input. For $n = 10$, $O(n^2)$ means iterating through $\binom{10}{2} = 45$ pairs of points, which is computationally trivial.
Some optimizations: 1. **Avoid validating duplicate rectangles** 2. **Early exit for invalid cases** 3. **Efficient point validation**
Given the small input size constraint, the $O(n^2)$ approach with these optimizations is efficient enough.

**Human:** How do you tell if something is in the rectangle again? What would a helper function look like?

**Model:** *[Abbreviated]*
To determine if a point is inside or on the boundary of a rectangle, you can use the rectangle's bounding coordinates. If the rectangle is defined by its diagonal corners $(x_1, y_1)$ and $(x_2, y_2)$, you can derive the rectangle's boundaries as $x_{min} = \min(x_1, x_2)$, $x_{max} = \max(x_1, x_2)$, $y_{min} = \min(y_1, y_2)$, $y_{max} = \max(y_1, y_2)$.
A point $(x, y)$ is inside or on the boundary if: $x_{min} \leq x \leq x_{max}$ and $y_{min} \leq y \leq y_{max}$.

*[User submits several iterations of code, eventually arriving at a correct solution]*

"Although the solution offered by the AI was one that I had originally thought of and most people would have thought of, it was still helpful in that it wasn't giving FALSE information. It basically just confirmed what I was thinking, and was helpful in that it confirmed the runtime wasn't able to reduced much further. Furthermore, I feel that if I was more thorough in asking about solving the problem in a really short time, it would give me more answers."

Figure 24: Interaction between human and AI model (GPT-4o) on finding the maximum area rectangle with point constraints. The model guides the user through an $O(n^3)$ approach by checking if points form valid rectangles, with no additional points inside or on the boundaries.

**Human:** how should i approach this

**Model:** *[Abbreviated]*
# Approach for Analyzing the Recurrence Relation $u_{k+1} = 2u_k - 2u_k^2$
1. **Find the Limit (L):**
* Assume $u_k \to L$ and $u_{k+1} \to L$ as $k \to \infty$
* Substitute into the recurrence: $L = 2L - 2L^2$
* Solve for L to find possible limits

2. **Determine the Actual Limit:**
* Calculate first few terms starting with $u_0 = 1/4$
* Analyze the function $f(x) = 2x - 2x^2$ and its derivative at fixed points

3. **Analyze Convergence Rate:**
* Consider transformation $v_k = 1 - 2u_k$
* Find recurrence for $v_{k+1}$ in terms of $v_k$
* Use the transformed sequence to find closed-form for $v_k$

4. **Translate the Condition:**
* Express $|u_k - L| \leq \frac{1}{2^{1000}}$ in terms of $v_k$
* Solve for the smallest $k$ satisfying this inequality

**Human:** what do you think are the first few terms of the sequence and how fast does it generally converge

**Model:** *[Abbreviated]*
# Exploring the Sequence Terms and Convergence Speed
1. **Calculating Terms:**
* Start with $u_0 = \frac{1}{4}$
* $u_1 = 2u_0 - 2u_0^2 = 2 \cdot \frac{1}{4} - 2 \cdot \left(\frac{1}{4}\right)^2$
* Continue for $u_2, u_3, \ldots$

2. **Observing Convergence:**
* Compare terms to potential limits
* Analyze how $|u_k - L|$ changes with each step

3. **Relating to Transformation:**
* Calculate $v_k = 1 - 2u_k$ for first few terms
* Examine the relationship between $v_{k+1}$ and $v_k$
* Determine how convergence of $v_k$ relates to convergence of $u_k$

**Human:** it converges exponentially fast, not linearly or quadratic.

*[User submits answer: 10]*
*[Answer is correct]*

"it gave me general intuition how to solve it but didn't say much other than just guess and check and find a pattern."

Figure 25: Interaction between human and AI model (Gemini 2.5 Pro) on a recurrence relation problem. The model outlines a systematic approach focusing on finding limits and transformation techniques. The user identifies the exponential convergence pattern and determines the correct answer.

Minimizing Array Changes to Make Differences Equal

**Human:** I'm assuming this can be done pretty fast, considering we can use k as somewhat like a boundary?

**Model:** *[Abbreviated]*
One key insight is that you only need to ensure each pair (i, n - i - 1) has the same absolute difference. Since you can replace any element with an integer from 0 to k, you have a fixed range for potential targets.

**Solution Strategy:**
1. Recognize that the final array must have pairs whose difference is some integer X ($<=$ k).
2. For each pair (i, n - i - 1), determine which X values can be achieved with 0, 1, or 2 changes.
3. Find the X that leads to the minimum total changes across all pairs.

**Human:** Can I have two pointers here?

**Model:** Yes. You could have one pointer at i = 0 and another at j = n-1, moving inward. At each step, you analyze how many replacements would be needed to make their difference match a candidate X.

**Human:** Can we not just count the amount of pairs with the same difference?

**Model:** *[Abbreviated]*
Simply counting the most common current difference can fail because you might achieve a different uniform difference with fewer total changes. You need to consider how many changes would be required for each pair to achieve each potential difference X, then choose the X requiring minimal changes overall.

**Human:** How do I know how many possible X's there are?

**Model:** All possible values of X lie in the set {0, 1, 2, ..., k}, since any replacements must be in [0, k].

**Human:** So this time won't be O(n)?

**Model:** *[Abbreviated]*
A naïve approach could lead to O(n·k) time, which is too large when n and k can both reach $10^5$. Instead, for each pair, determine which differences can be achieved with 0, 1, or 2 changes, storing this information compactly. Then find the difference X that minimizes total changes.

**Human:** So what are we storing?

**Model:** For each pair, store: - The single difference achievable with 0 changes (current absolute difference) - A compact representation of differences achievable with 1 change - By default, other differences in [0, k] require 2 changes

*[User attempts multiple implementations, encountering various errors]*

**Human:** *[Final implementation attempt, which times out]*

---

"Although this AI 'seemed' smarter than the others in that it was giving me better solutions, the overall explanation was confusing such that I wasn't able to understand it! There were points where the grammar was interesting and it would throw me off. If the grammar usage was a bit better I could've implemented what it was asking me, but I started getting frustrated trying to understand what it was trying to say."

Figure 26: Interaction between human and AI model (Deepseek-V3) on a problem about minimizing array modifications to equalize differences. Despite a theoretically sound approach, communication barriers prevented successful implementation.

**Human:** how should i approach this problem

**Model:** *[Abbreviated]*
# Approach to the Problem
1. **Understand the Properties of z:** You have $z = a + bi$, where $a$ and $b$ are integers.
2. **Analyze the Modulus Condition:** Given $|z| = \sqrt{p}$ where $|z| = \sqrt{a^2 + b^2}$, we get $a^2 + b^2 = p$.
3. **Properties of p:** Consider which prime numbers can be expressed as $a^2 + b^2$.
4. **Calculate $z^3$:** Expand $(a + bi)^3$.
5. **Identify Real and Imaginary Parts of $z^3$:** Find $Re(z^3)$ and $Im(z^3)$.

**Human:** explain step 4 in more detail

**Model:** *[Abbreviated]*
You need to compute $z^3 = (a + bi)^3$ using the binomial expansion: $(a + bi)^3 = a^3 + 3a^2(bi) + 3a(bi)^2 + (bi)^3$
Remember that: - $i^2 = -1$ - $i^3 = -i$
After substituting, you can rewrite the expression and group the real and imaginary parts: $z^3 = (a^3 - 3ab^2) + (3a^2b - b^3)i$
So $Re(z^3) = a^3 - 3ab^2$ and $Im(z^3) = 3a^2b - b^3$.
**Regarding the Triangle Inequality:** The condition requires that $p$, $Re(z^3)$, and $Im(z^3)$ form a triangle, which means: - All side lengths must be positive - Any two sides must sum to more than the third side
**Human:** $Re(z^3) = a^3 - 3ab^2$ and $Im(z^3) = 3a^2b - b^3$

**Human:** explain step 6 in more detail

**Model:** *[Abbreviated]*
# Apply the Triangle Conditions
For a triangle with sides $p$, $s_2 = a^3 - 3ab^2$, and $s_3 = 3a^2b - b^3$:
1. **Positivity Conditions:** - $p > 0$ (always true since $p$ is prime) - $s_2 > 0$, which requires $a > \sqrt{3}b$ (assuming $a, b > 0$) - $s_3 > 0$, which follows if $a > \sqrt{3}b$
2. **Triangle Inequality:** - $p + s_2 > s_3$ - $p + s_3 > s_2$ - $s_2 + s_3 > p$
These simplify to: - $|a^3 - 3a^2b - 3ab^2 + b^3| < a^2 + b^2$ - $a^3 + 3a^2b - 3ab^2 - b^3 > a^2 + b^2$
**Strategy:** Find pairs of positive, coprime integers $(a, b)$ that satisfy: - $a, b$ are positive integers - $a$ and $b$ are coprime - $p = a^2 + b^2$ is prime and $p < 1000$ - $a > \sqrt{3}b$ - Triangle inequality conditions hold
You've found primes 17 (from $a = 4, b = 1$), 89 (from $a = 5, b = 8$), 349 (from $a = 18, b = 5$)...
*[User submits 199, which is incorrect, then 349, which is correct]*

"Helped me with my thought process especially with certain identities and tricks. I kind of forgot certain properties of complex numbers so the AI was pretty good at getting me back up to speed. Aside from a few glitches, I feel it did a pretty good job at explaining things and setting a framework with clear steps to take."

Figure 27: Interaction between human and AI model (Gemini 2.5 Pro) on a AIME problem requiring complex number manipulation. The model provides a step-by-step approach, helping the user navigate through mathematical derivations and systematically find the largest prime meeting all conditions.

