# OpenReview forum: "When Models Know More Than They Can Explain: Quantifying Knowledge Transfer in Human-AI Collaboration"
_NeurIPS.cc/2025/Conference — NeurIPS 2025 poster_

### Official Review · Reviewer_qEBD · 2025-06-10

**Clarity:** 3
**Significance:** 4
**Originality:** 3
**Rating:** 4
**Confidence:** 3

**Summary:**

This paper introduces a novel empirical framework to evaluate knowledge transfer from LLMs to humans in collaborative problem-solving settings on coding and mathematical reasoning tasks. The authors isolate and quantify the transfer of reasoning from LLMs to users through a two-phase user study involving 118 participants. Participants first ideate with an LLM, then independently solve the problem, allowing the authors to measure whether the reasoning of LLMs has been effectively internalized. They find that while model benchmark performance generally correlates with collaborative success, this relationship is inconsistent across LLMs and tasks, suggesting that knowledge transfer is a distinct capability requiring separate evaluation.

**Questions:**

See Weaknesses.

**Ethical Concerns:**

["NO or VERY MINOR ethics concerns only"]

**Final Justification:**

I will keep my score.

**Limitations:**

Yes.

**Paper Formatting Concerns:**

No.

**Quality:**

3

**Strengths And Weaknesses:**

Strengths:

* This paper is well-motivated for understanding how LLMs influence human performance in collaborative tasks. It addresses an underexplored aspect of human-AI interaction by investigating how LLMs can actually help humans become more capable, rather than just outsourcing tasks to AI.
* The authors propose a formal framework to model the projection of AI knowledge into human-understandable forms. This contributes to the theoretical understanding of communicative alignment and knowledge transfer between humans and machines.
* The proposed framework effectively isolates reasoning transfer, enabling performance gains to be cleanly attributed to internalised LLM knowledge.
* The analysis includes both quantitative metrics and qualitative clustering of interaction behaviors, offering novel insights into which strategies lead to successful knowledge transfer.

Weaknesses:

* All experiments are evaluated in zero-shot settings, which might understate the potential of LLMs for better pedagogical alignment. Could the authors report the ablation performance of human-AI interaction under the CoT setting, for better exploring the intrinsic assistive capabilities of LLMs? For example, the authors find that high-performing math models often produce dense symbolic expressions when assisting humans, which can be confusing for users. However, this can be easily addressed by modifying the prompt to encourage the LLM to focus more on helping the user rather than solving the problem independently.
* Although the authors propose an intuitive conceptual framework dividing knowledge into shared, AI-exclusive, and human-exclusive regions, they do not attempt to quantitatively analyze these segments independently. While they argue such divisions are difficult to measure, a subjective rating from participants on perceived knowledge novelty or conflict could offer insights into how often LLMs provide genuinely instructive concepts.

---

> ### Author Rebuttal · Authors · 2025-07-30
>
> Thank you for your thoughtful review and your recognition of our work. We address your concerns below:
>
> > For example, the authors find that high-performing math models often produce dense symbolic expressions when assisting humans, which can be confusing for users. However, this can be easily addressed by modifying the prompt to encourage the LLM to focus more on helping the user rather than solving the problem independently.
>
> This is a great point, and we fully agree that prompt design plays a critical role in the effectiveness of human-AI collaboration. However, we deliberately chose not to exhaustively optimize prompts at this stage of the investigation for multiple reasons:
>
> First: evaluating a broad set of prompting strategies combinatorially across 8 models and multiple skill hierarchies quickly becomes infeasible. For each additional prompt variant, we would require a proportional increase in participant data to preserve statistical power and control for interaction effects. Given the resource-intensive nature of our user studies, with 118 skilled humans already involved, scaling to cover a full prompt × model × task grid would have imposed prohibitive data collection and analysis costs.
>
> Instead, we adopt a *top-down* approach: we evaluate models under minimal guidance in order to measure their *intrinsic* capacity to communicate and convey reasoning to humans. Our goal is not to optimize each model for success via prompt engineering, but rather to assess how effectively a model, as-is, can adaptively support a user when given agency in presenting its knowledge. This reflects a more ecologically valid setting where users are not necessarily prompt engineers, and where the quality of support depends on the model’s communicative competence, not on hand-tuned scaffolding. We believe this setup also aligns more closely with the real-world deployment scenario (as mentioned in Section 1) we hope to understand.
>
> Furthermore, evaluating alternate prompting strategies *post hoc* risks introducing bias and overfitting to our task suite. It would allow us to “cherry-pick” effective prompts retrospectively, creating an illusion of pedagogical alignment that might not generalize beyond our specific test cases. By contrast, our current approach generalizes and enables us to surface naturally occurring shortcomings in LLM communication, such as the overuse of dense symbolic math, based in genuine participant confusion.
>
> Our qualitative clustering of interaction behaviors and participant feedback in Section 6 already gives us valuable insights into what kinds of LLM strategies are effective or counterproductive. These empirical observations can inform future work that systematically explores prompt design with a narrowed scope, guided by evidence rather than searching from the get go in a large solution space. Our analysis could further inform model builders aiming to improve the basic collaboration skill of LMs.
>
> > Although the authors propose an intuitive conceptual framework dividing knowledge into shared, AI-exclusive, and human-exclusive regions, they do not attempt to quantitatively analyze these segments independently. While they argue such divisions are difficult to measure, a subjective rating from participants on perceived knowledge novelty or conflict could offer insights into how often LLMs provide genuinely instructive concepts.
>
>
> Absolutely. We aim to provide such a measurement in section 6, where we aggregate open-ended participant feedback to shed light on specific interaction mechanisms under the umbrella of our framework in Section 3, tagged with their frequency and correlation with success.
>
>
> Thank you again for your thoughtful review. We hope this additional context resolves your lingering concerns. We look forward to continuing the conversation to improve your perception of our work and clear up any additional questions!

---

### Official Review · Reviewer_M7Bb · 2025-06-25

**Clarity:** 3
**Significance:** 4
**Originality:** 4
**Rating:** 4
**Confidence:** 4

**Summary:**

The paper introduces a conceptual and experimental framework for quantifying how effectively LLMs teach humans. In a study with 118 participants and 8 models, it measures knowledge transfer on coding and math tasks using objective solve rates and subjective ratings.

**Questions:**

1. Can you plot knowledge-transfer gain against each participant’s pre-test score (or self-reported experience) to reveal whether low- or high-expertise users benefit more from LLM collaboration?
2. You report scaffolding boosts low-skill users but bores high-skill ones. Could you sketch how an adaptive strategies such as concise strategy for experts, detailed steps for novices, might reduce this gap, perhaps via a brief analysis or simulation?

**Ethical Concerns:**

["NO or VERY MINOR ethics concerns only"]

**Final Justification:**

The knowledge-transfer gain information provided in the rebuttal serves as a valuable point of reference for understanding how effectively LLMs can transfer knowledge to humans, thereby enhancing the significance of the paper.

**Limitations:**

Yes.

**Paper Formatting Concerns:**

The paper is well-formatted.

**Quality:**

3

**Strengths And Weaknesses:**

#### **Strengths**
1. Shifts focus from model accuracy to AI-to-human knowledge transfer, opening a fresh research axis.
2. 118 participants, eight LLMs, clear two-phase protocol; data, code, and dashboard are released for reproducibility.
3. Identifies explanation structure and scaffolding style as key drivers of successful teaching, guiding future alignment work.

#### **Weaknesses**
1. Evaluation is confined to coding and math, leaving knowledge-transfer effectiveness in non-STEM domains unvalidated.
2. The study does not examine how varying prompt styles or settings influences the models’ teaching effectiveness.

---

> ### Author Rebuttal · Authors · 2025-07-30
>
> Thank you so much for your thoughtful review and your recognition of our work. We aim to address your core concerns below:
>
> > Evaluation is confined to coding and math, leaving knowledge-transfer effectiveness in non-STEM domains unvalidated.
>
> We agree that this is a limitation. However, we decided to constrain our study to math and coding for 2 reasons. First, math and coding are uniquely suited to automatic verification while still having reasoning-intensive, open-ended solutions. It’s relatively much more expensive and difficult to reliably grade participants’ answers for non-STEM fields. Second, these domains match our methodology exceptionally well since “solving” a coding or math problem often requires understanding something concrete about an algorithm or proof that can be more reliably tested than domains that may have more unstructured solutions. While we agree that evaluating more diverse domains, we see our contribution as providing not just an analysis of knowledge transfer, but also a general methodology that can be applied to additional domains. We hope that this contribution can help promote the study of knowledge transfer in other domains as future work.
>
>
> > The study does not examine how varying prompt styles or settings influences the models’ teaching effectiveness.
>
> This is a good point, and we fully agree that prompt design plays a critical role in the effectiveness of human-AI collaboration. However, we deliberately chose not to exhaustively optimize prompts at this stage of the investigation for multiple reasons:
>
> First: evaluating a broad set of prompting strategies combinatorially across 8 models and multiple skill hierarchies quickly becomes infeasible. For each additional prompt variant, we would require a proportional increase in participant data to preserve statistical power and control for interaction effects. Given the resource-intensive nature of our user studies, with 118 skilled humans already involved, scaling to cover a full prompt × model × task grid would have imposed prohibitive data collection and analysis costs.
>
> Instead, we adopt a *top-down* approach: we evaluate models under minimal guidance in order to measure their *intrinsic* capacity to communicate and convey reasoning to humans. Our goal is not to optimize each model for success via prompt engineering, but rather to assess how effectively a model, as-is, can adaptively support a user when given agency in presenting its knowledge. This reflects a more ecologically valid setting where users are not necessarily prompt engineers, and where the quality of support depends on the model’s communicative competence, not on hand-tuned scaffolding. We believe this setup also aligns more closely with the real-world deployment scenario (as mentioned in Section 1) we hope to understand.
>
> > Can you plot knowledge-transfer gain against each participant’s pre-test score (or self-reported experience) to reveal whether low- or high-expertise users benefit more from LLM collaboration?
>
>
> This is a great point. We have added this in the Appendix of our paper. To summarize the findings, we find it to be roughly even across skill levels, with a slight dip towards higher skilled levels (for each model).
>
> > You report scaffolding boosts low-skill users but bores high-skill ones. Could you sketch how an adaptive strategies such as concise strategy for experts, detailed steps for novices, might reduce this gap, perhaps via a brief analysis or simulation?
>
> This is an interesting point that would be good for future work! Because these are strategies we’ve observed post-hoc, we leave experiments on these strategies to future work. A potential method for evaluating this is to utilize the same general framework our paper provides to evaluate collaborative effectiveness and also provide models pre-hoc with users’ ELO in order to contextualize model responses. We may use models that are explicitly instructed to utilize/not to utilize scaffolding as baselines.
>
>
> Hopefully this additional context resolves your concerns related to domain breadth and prompting methodology. We look forward to continuing the conversation to improve your perception of our work and clear up any additional questions!

---

> > ### Comment · Reviewer_M7Bb · 2025-08-02
> >
> > Thank you for the authors' responses. The focus on math and coding domains is understandable, and the rationale for not optimizing prompts is convincing.
> >
> > Although the revised paper will include knowledge-transfer gains relative to each participant’s pre-test score, would it be possible to present this information here in a tabular or other structured format for the reviewers’ reference?

---

> ### Author Response · Authors · 2025-08-03
>
> Glad we were able to clear up your previous concerns.
>
> > Although the revised paper will include knowledge-transfer gains relative to each participant’s pre-test score, would it be possible to present this information here in a tabular or other structured format for the reviewers’ reference?
>
> Yes, we can absolutely present this information here. Because every task was calibrated to lie one fixed Elo step above each participant’s own Elo rating, and our Elo ratings are fitted with a Bradley–Terry logistic model, we can analytically convert that fixed Elo gap into an expected unaided (“H”) success probability, 14.2 % for code and 8.1 % for math, without requiring the participant to solve additional baseline problems. Collaboration accuracy (“H + M”) therefore already measures how much additional knowledge (knowledge transfer) each model contributes. For completeness, we now present these data in tabular form in Table 2, explicitly showing the solo baseline and subtracting it from every collaborative score.
>
> | Setting                        | GPT-4.1 | GPT-4o | o1  | GPT-4.5 | DS-V3 | Llama-4 | Cld-3.7 | Gem-2.5 |
> |--------------------------------|---------|--------|-----|---------|-------|---------|---------|---------|
> | **Code (H)**                   | 14.2    | 14.2   | 14.2 | 14.2    | 14.2  | 14.2    | 14.2    | 14.2    |
> | **Code (M)**                   | 68.8    | 16.7   | 55.0 | 68.4    | 33.3  | 47.1    | 45.0    | 81.3    |
> | **Code (H + M)**               | 65.0    | 40.0   | 55.0 | 69.0    | 51.9  | 54.7    | 70.0    | 71.3    |
> | **Code ((H + M) – H)**         | +50.8   | +25.8  | +40.8 | +54.8   | +37.7 | +40.5   | +55.8   | +57.1 |
> | **Code ((H + M) – M)**         | -3.8    | +23.3  | +0.0 | +0.6    | +18.6 | +7.6    | +25.0 | -10.0   |
> | **Math (H)**                   | 8.1     | 8.1    | 8.1  | 8.1     | 8.1   | 8.1     | 8.1     | 8.1     |
> | **Math (M)**                   | 47.6    | 8.3    | 83.3 | 33.3    | 46.2  | 47.8    | 20.8    | 68.4    |
> | **Math (H + M)**               | 75.7    | 56.7   | 85.8 | 60.8    | 70.8  | 72.6    | 81.7    | 79.5    |
> | **Math ((H + M) – H)**         | +67.6   | +48.6  | +77.7 | +52.7 | +62.7 | +64.5  | +73.6   | +71.4   |
> | **Math ((H + M) – M)**         | +28.1   | +48.4  | +2.5 | +27.5   | +24.6 | +24.8   | +60.9 | +11.1   |
>
> In fact, one of the paper’s key contributions is the above experimental framework for quantifying knowledge transfer. By leveraging a Bradley–Terry Elo calibration, we can predict each participant’s solo success probability for any task directly from its difficulty rating. This means we no longer need participants to tackle separate benchmark problems to establish a baseline; the model supplies that estimate “for free.” As a result, we obtain an accurate measure of human-only performance without extra trials, making the study both sample-efficient and cost-effective while still yielding a calibrated knowledge-transfer signal.
>
> Hope this was helpful! If you have any other questions, concerns, or requests that would help us improve your understanding and perception of our work, we would be happy to provide.

---

> ### Comment · Reviewer_M7Bb · 2025-08-03
>
> Thank you for including this critical piece of information, which helps readers to understand how well the knowledge is transferred to human. Please make sure to also include the this updated table in your paper. I will raise the significance score while maintaining my current rating, with a slight inclination towards acceptance.

---

### Official Review · Reviewer_zabu · 2025-07-01

**Clarity:** 3
**Significance:** 3
**Originality:** 4
**Rating:** 5
**Confidence:** 4

**Summary:**

This paper studies the knowledge transfer from LLMs to humans via a large-scale user study with 118 participants. In a two-phase setup, the authors aim to measure the influence of model reasoning on human understand. In the first phase, humans can ideate with an LLM on problem-solving strategies on math and coding tasks, while in the second phase the human independently implement the solutions. The authors find that model benchmark performance correlates with collaborative outcomes, but that this relationship is inconsistent and contains several outliers. Additionally, the results show that model solo performance correlates positively with knowledge transfer, while the relationship between human preferences is mixed (positive for code, negative for math). The paper argues that this points to knowledge transfer from LLMs to humans is a distinct capability which may require dedicated optimization.

**Questions:**

- Could the authors show the distribution of some qualitative findings across models? For example, do models like Gemini-2.5-Pro fail in knowledge transfer due to "style issues"?
- Do the authors see some relationship with respect to the recent literature on the impact of LLMs on education? Where the general trend is, if LLMs help but do not give the final response, students have a faster learning rate?

**Ethical Concerns:**

["NO or VERY MINOR ethics concerns only"]

**Final Justification:**

The rebuttal did not change my acceptance recommendation.

**Limitations:**

The authors acknowledge and discuss limitations of their work in the main part of the paper.

**Paper Formatting Concerns:**

No formatting concerns.

**Quality:**

4

**Strengths And Weaknesses:**

Strengths:
- A very well executed paper where the main effects are robust with respect to some confounders, and the additional qualitative analysis helps to understand the quantitative findings.
- The two-phase setup allows for a clean isolation of the knowledge transfer effect.
- Interesting take on knowledge transfer being a distinct capability, and interesting to see that the slope <1 when compared to solo model performance.

Weaknesses:
- Some seemingly "causal" claims are only the result of an observational study. Randomising (or controlling for) the response-style of LLMs may help to make some claims (such as the effect of adaptive scaffolding) stronger.

---

> ### Author Rebuttal · Authors · 2025-07-30
>
> Thank you for your positive review of our paper! We are glad to hear you find our experimental setup “well executed” and our results interesting as well. We aim address your remaining concerns below:
>
> > Randomising (or controlling for) the response-style of LLMs may help to make some claims (such as the effect of adaptive scaffolding) stronger.
>
> We agree, but we do see a give and take here. We believe that the response-style is a choice that the LLMs themselves should be able to make, as an LLM’s ability to select how to convey information is an important factor in its knowledge transfer abilities. Although it may make some claims stronger, controlling for the response style would remove this component, which may limit the generalizability of our study.
>
> > Could the authors show the distribution of some qualitative findings across models? For example, do models like Gemini-2.5-Pro fail in knowledge transfer due to "style issues"?
>
> Yes: we do include a small amount of analysis on this in Section 6. We don’t dive into it in too much detail because the *qualitative* findings are only really able to be analyzed through participants that provided actual feedback after completing a problem: which is not all of the interactions. Therefore, such a metric would be skewed in the types of participants (via skill level) that it represents, which may paint an inaccurate picture.
>
> > Do the authors see some relationship with respect to the recent literature on the impact of LLMs on education?
>
> Absolutely. Along with the recent results assessing the cognitive debt of LLM usage on writing tasks and learning [1] , we believe it to be incredibly important to properly isolate and evaluate the ability of humans to properly internalize LLM reasoning, as well as LLM ability to properly express reasoning in understandable terms. This knowledge transfer is core to the utility of LLMs in educative contexts: yet despite this, few works have tackled this question.
>
> Hopefully this additional context resolves your lingering questions. Thank you again for your recognition of our work!
>
> References:
> [1] Kosmyna, N., Hauptmann, E., Yuan, Y. T., Situ, J., Liao, X. H., Beresnitzky, A. V., ... & Maes, P. (2025). Your brain on chatgpt: Accumulation of cognitive debt when using an ai assistant for essay writing task. arXiv preprint arXiv:2506.08872.

---

> > ### Comment · Reviewer_zabu · 2025-08-04
> >
> > Thank you for the rebuttal. I agree with the trade-off by controlling for response style, however, I would state this somewhere in the paper. In future work, controlling for this (e.g., via prompting or fine-tuning) may reveal that the late-stage RL fine-tuning of some models does not optimize for knowledge transfer. (You already state this partially somewhere in the paper). Additionally, I believe that making the connection with the (small) literature on LLMs and education could be beneficial for the paper. I keep my recommendation on acceptance.

---

### Official Review · Reviewer_aoxv · 2025-07-05

**Clarity:** 2
**Significance:** 3
**Originality:** 3
**Rating:** 3
**Confidence:** 4

**Summary:**

This paper investigates a well-posed and highly relevant question: does a large language model's internal reasoning capability directly translate into effective knowledge transfer during human-AI collaboration? As I understand it, the authors propose a conceptual framework to distinguish between model-exclusive, human-exclusive, and shared knowledge. They test this with a two-phase experiment where participants first collaborate with an LLM for ideation on coding and math problems, and then must solve the problem independently. This second phase is designed to be a direct test of how well knowledge was actually transferred. The work aims to quantify this transfer process and identify the behavioral and strategic factors that lead to successful collaboration, independent of a model's raw performance.

**Questions:**

My main concern is the experimental setup. Without a human-human control group, how can you be certain that your findings are specific to Human-AI knowledge transfer, rather than general principles of how these participants collaborate or learn from any partner? Addressing this would significantly strengthen the claims.

Could you provide a clearer rationale for several key experimental design choices? Specifically:

Why were participants forbidden from taking notes? Prohibiting participants from taking notes seems counter-intuitive and may unintentionally shift the evaluation towards rote memorization rather than the intended measure of knowledge transfer.

Why prevent participants from accessing the AI during the implementation phase? The rationale for this constraint is unclear and should be justified.

What varies between the three zero-shot attempts for measuring model skill?

Why the different submission attempt limits (10 for code, 5 for math)?

Your interpretation of results seems problematic in places. Interpreting Gemini-2.5-Pro's performance as "reduced collaborative efficacy" is potentially misleading. It is more likely a demonstration of a ceiling effect, where models with high initial solo performance have an inherently smaller margin for improvement, rather than a failure of the collaboration itself. Could you discuss this alternative explanation?

Some findings may be artifacts of experimental constraints. For example, was the preference for formal notation in math observed even when users explicitly asked for simpler methods? Could this effect be hampered by the protocol's prohibition of pseudocode, which is often a much more accessible form of communication?

Should participants know which model they're paired with, and if not, were they able to infer the model's identity?

The choice of uniform temperature of 0.7 requires justification, especially since the assertion that this reflects how users "commonly interact" is debatable when using API access versus web interfaces with specialized system prompts.

You mention the bidirectional process of humans projecting reasoning to the model, which you note is tested qualitatively. I understand this is hard to measure, but I'd like to hear the authors' ideas on how this could be approached quantitatively in future work.

**Ethical Concerns:**

["NO or VERY MINOR ethics concerns only"]

**Limitations:**

The authors do not adequately address the limitations of their work. A key limitation that should be discussed is the lack of a human-human control group, which makes it difficult to isolate the effects of AI-specific interaction. Additionally, the study's constraints such as prohibiting note-taking and pseudocode are significant limitations that may impact the external validity of the findings regarding naturalistic knowledge transfer, and this should be acknowledged.

**Paper Formatting Concerns:**

See above (minor)

**Quality:**

2

**Strengths And Weaknesses:**

The paper's primary strength is that it is framed around a well-posed and highly relevant research question. The attempt to quantify knowledge transfer as a distinct capability from model performance is a significant and original contribution to the field of human-AI collaboration. The experimental design has several strong components, particularly the two-step process for estimating human skill and the method of calibrating task difficulty to be just above the participant's level, which ensures collaboration is meaningful.

However, the paper has several weaknesses, primarily in its experimental design and clarity.

Quality: My main concern is the lack of a human-human control group. Without this baseline, it's difficult to determine if the observed effects are specific to AI interaction or simply reflect general collaborative abilities. Furthermore, some interpretations are questionable; the claim of "diminishing returns" for high-performing models seems more likely to be a simple ceiling effect, where there's less room for improvement.

Clarity: Several design choices are not clearly justified. It's unclear why participants were forbidden from taking notes (this seems to test for fact memorization unintentionally), why submission attempt limits differ between tasks (10 vs. 5), or why a uniform temperature of 0.7 was chosen for all models, especially given that some model providers recommend different defaults and the non-creative nature of these tasks. The conceptual framework diagram in Figure 2 is frankly not a helpful aid and does little to improve the reader's understanding. Table captions are also not descriptive enough to be understood independently, especially for newly coined terms like HTM, HMT, and MHT. This is an example of a broader issue where some experimental design choices feel arbitrary and are left unexplained, which creates confusion and detracts from the paper's otherwise strong ideas.

Significance: The qualitative analysis is well-executed; however, its contribution should be clarified by explicitly highlighting which findings are expected versus unexpected. The paper should frame its results in the context of existing literature, noting where they corroborate prior work and where they represent novel discoveries.

---

> ### Author Rebuttal · Authors · 2025-07-30
>
> Thank you for your thoughtful review! We address your concerns below:
>
> > Without a human-human control group, how can you be certain that your findings are specific to Human-AI knowledge transfer…
>
> Below we clarify why a human-human control group, while interesting independently, is not essential for the claims made in this paper.
>
> 1. **Research focus and claim scope**
>
> Our goal is to measure the relationship between LLMs’ solo capabilities and their ability to transfer / communicate their knowledge to humans as these abilities improve. We believe that understanding this relationship is primarily important because LLMs aren’t human, so as we cede more responsibilities and privileges to LLMs, it is critical that we’re capable of overseeing and understanding *their* reasoning, rationales, and behaviors. Therefore, all statistical tests, figures, and conclusions in the paper are framed around relative differences between models’ knowledge transfer capabilities, so a human‑human condition would not change any of the relative rankings that constitute our core quantitative contributions. Additionally, we do not make any claims as to whether the interactions observed in Section 6 apply *only* to Human-AI collaboration: they could be (and most likely are) general collaboration paradigms that could also apply to Human-Human collaboration. Thus, not having a human-human condition does not discount any of our findings. We do acknowledge that providing a comparison to Human-Human collaboration mechanisms could be a very interesting auxiliary result that links our work to existing work in Human-Human collaboration. We are excited to explore this in the future.
>
>
> 2. **Generic collaboration benefits are controlled for implicitly**
>
> Any generic boost from working with a partner, be it human or AI, acts as a constant offset that applies equally to every LLM condition. Because participants were randomly assigned to one of eight models, these generic effects are evenly distributed and cannot explain the systematic, model‑specific differences in our findings. In fact, our mixed‑effects logistic regression (Sec. 5, App. C.3) shows that model identity remains a highly significant predictor (p < 0.001) of post‑collaboration success even after controlling for user skill, task type, interaction length, and LLM familiarity. This suggests that the effects we report are tied to which AI collaborator is present, not merely to collaboration per se.
>
>
> 3. **Methodological and logistical considerations**
>
> Including a human-human baseline would likely also have required significantly more resources and demands on our human participants than we had available, diverting from the core model focused factors we aimed to study.
>
>
> To your point, we have revised our paper to ensure this point is clear, including clarifying in Section 4.1 why our hypotheses and analyses do not rely on such a baseline, as well as guiding readers to related literature in the related work that reports human‑human transfer rates.
>
> > Rationale on prohibiting note taking and pseudocode, and prohibiting participants from accessing the AI during the implementation phase
>
> The central outcome variable in our study is *knowledge that participants have actually internalized* after collaborating with an LLM. Allowing external memory aids (hand‑written notes, screenshots of pseudocode, or live re‑queries to the model) would blur that signal by letting participants outsource recall rather than encode and retrieve the material themselves. In pilot sessions we observed exactly this behaviour: if participants could jot down or copy‑paste an answer scaffold, many stopped asking clarifying questions and could perform on the post‑task even when they clearly had not grasped the underlying concept.
>
> The manipulation follows the same logic as a closed‑book exam in educational research: retrieval practice without aids is a stronger indicator of true learning than recognition with aids. Classic work on the *testing effect* [1] shows that removing external prompts yields better long‑term retention and cleaner measurement of what was learned. Our prohibition of pseudocode plays a parallel role in math tasks, where a line‑by‑line algorithm can act as a de‑facto crib sheet.
>
> Note that these restrictions were uniform for all eight model conditions. Therefore they also cannot explain the model‑specific performance differences we report; at worst they lower absolute scores equally, leaving relative ordering intact. To your point, we have added a note in the limitations section acknowledging that real‑world programming often is “open‑book.” A useful next step is to replicate our framework in an open‑resource setting and compare retention under different aid regimes. We believe that our current closed‑book design, however, is the necessary first step to quantify transfer in isolation.
>
> > What varies between the three zero-shot attempts for measuring model skill?
>
> Because at test time during collaboration we sample at non-zero temperature, therefore, we only provide humans that the model repeatedly gets wrong to reduce variance and to ensure that at test time, collaboration is meaningful. This creates the need to sample multiple times to determine this.
>
> > Why the different submission attempt limits (10 for code, 5 for math)?
>
> This is because the space of possible answers for code (imagine a code function) is much larger than math, which is often confined to integers from 1-1000 for AIME questions. Empirically during our initial testing we found that giving too many attempts for math allowed for a small, but non-zero chance of completely guessing the answer correctly. Additionally, for math, we observed that participants tended to not utilize additional attempts past ~5 attempts effectively (i.e. random guessing behavior arose). These numbers were determined through post-user surveys + manually inspecting solve sessions in a testing phase before the actual experiment. Our rationale for design decisions based on our trial experiments were not added to the paper and this was definitely an oversight: we have now added this to appendix section C. We really appreciate your pointing out details that you find unclear!
>
> >  It is more likely a demonstration of a ceiling effect, where models with high initial solo performance have an inherently smaller margin for improvement, rather than a failure of the collaboration itself.
>
> We agree that there could be ceiling effect: however, we also observe that Claude-3.7-sonnet is able to exceed Gemini’s ability in inducing collaborative performance in math and closely matching it in code, despite only having half and a third of Gemini's solo performance, respectively (Section 5). Our description of Gemini’s reduced collaborative efficacy thus more points to the fact that despite its high performance, it was not able to output a proportionally higher performance in humans. We don’t discount the possibility of a ceiling effect: but it’s hard to say either way without more data points with higher model performance, which was not possible at the time as Gemini represented the highest performance in our tested benchmark and there were simply no better models then. Nonetheless we think this is a great nuance to include, and we have added this in our discussion of results in Section 5.
>
> >  For example, was the preference for formal notation in math observed even when users explicitly asked for simpler methods? Could this effect be hampered by the protocol's prohibition of pseudocode, which is often a much more accessible form of communication?
>
> As discussed previously, we must prohibit pseudocode to ensure our experimental procedure accurate measures *internalized knowledge*. We find that this constraint does **not** force models into dense symbolic exposition; they were free to, and often did, answer in ordinary prose. In our logs we see many exchanges where participants explicitly requested “plain English” or “a simpler explanation,” and the LLM immediately shifted style, because the guard‑prompt only disallowed verbatim algorithmic scaffolds, not natural‑language reasoning. Hence the occasional preference for formal notation arises from model defaults and user choices, not from any protocol.
>
> > Should participants know which model they're paired with, and if not, were they able to infer the model's identity?
>
> Participants were not informed which model they were paired with. We also did not observe any attempts by participants to identify the model through prompting or other means.
>
> > The choice of uniform temperature of 0.7 requires justification, especially since the assertion that this reflects how users "commonly interact" is debatable…
>
> For sure. We fixed the temperature to eliminate temperature as a confounder when comparing knowledge‑transfer effectiveness. We chose 0.7 because it is commonly recommended by providers (ie OpenAI in their API docs), balances diversity and determinism, and unlike model‑specific web presets, can be applied uniformly through the API. Because we lack visibility into the proprietary system prompts that shape web‑interface behaviour, controlling this parameter ensures a level playing field and keeps observed differences attributable to the models themselves rather than to decoding settings.
>
> Hopefully this additional context resolves your concerns. Thank you genuinely for your detailed review: reading through your review was invaluable to helping us improve our paper based upon your suggestions. We have already made many clarifying additions in response to your feedback. We believe strongly in our work, so we look forward to continuing the conversation to improve your perception of our work and clear up any additional questions!
>
> References:
>
> [1] Roediger, H. L. III, & Karpicke, J. D. (2006). Test-enhanced learning: Taking memory tests improves long-term retention. Psychological Science, 17(3), 249–255.

---

### Official Review · Reviewer_wnSo · 2025-07-07

**Clarity:** 3
**Significance:** 3
**Originality:** 4
**Rating:** 5
**Confidence:** 3

**Summary:**

This paper investigates knowledge transfer in Human-AI collaboration and seeks to quantify this. In the abstract, it states “... it is crucial that they [LMMs] express their reasoning in ways humans can understand and learn from. However, this capability remains relatively less understood and under-evaluated.” Therefore, the paper reports on a study of human-AI knowledge transfer with 118 human participants that relies on programming tasks from LiveCodeBench and competition math problems.

The experiment has two phases: 1) first, humans and AI collaborate to solve the tasks, this is an ideation phase where the problem and solution are explored, and 2) in the second step, the humans must solve the task alone, only based on the understanding gained from the first phase.

In order to ensure knowledge transfer, the idea is to give humans tasks that are just above their expertise level (as measured by ELO). The idea is that if humans are capable of solving these tasks, then they have solved a task outside their measured capability level and thus learned from the first phase in which they collaborated with AI.

The results (from the conclusion): model performance generally correlates with collaborative outcomes, this is inconsistent. Interaction mechanisms that help explain these gaps are investigated (Figure 4).

**Questions:**

Please see strengths and weaknesses.

**Ethical Concerns:**

["NO or VERY MINOR ethics concerns only"]

**Final Justification:**

The authors stilled my concerns, and improved their explanations. The research is very interesting and I hope to see it published, so I have updated my recommendation to Accept.

**Quality:**

3

**Strengths And Weaknesses:**

## SUMMARY
This paper is intriguing, as it seeks to tackle a very important and interesting problem related to how humans can learn from AI instead of just blindly relying on the model output. An MIT study (Kosmyna et al., 2025) found that cognitive activity scaled down when relying on external tools, such as Google, but to a larger extent LLMs. The research presented in this paper sets up an experiment where the goal instead is for humans to learn from LLMs and apply this knowledge in solving problems, which in my opinion suggests a much better way to introduce LLMs to human problem solving and decision making.

However, the paper fails to answer the very interesting questions that it seeks to answer by analyzing and interpreting the results in a way that does not reach its full potential. I am not sure whether the experiments, as they were implemented, actually could test the hypotheses that were posed and thus whether they are able to answer the research questions they seek to answer. So, I am interested in understanding this.

The potential of this research is very high, but the execution does not capture the full potential - yet.

## STRENGTHS
There are several things I like with this paper:
* The topic is interesting, highly relevant and timely. The consequences of releasing highly capable LLMs into the society are not properly understood, but many problems are appearing. People rely on results from LLMs without understanding or quality assuring them. This is problematic. Hence, finding out whether and how we - as human beings – can grow our capabilities by collaborating with them is not only enticing but also very valuable. LLMs could unlock the potential for not only personalized learning but for solving any task. The way the paper is framed – in my understanding - is exactly to do this.
* A central point in the experiment design is that it captures task toughness, and human and AI capabilities in the form of ELO scores. This enables the researchers to control how challenging tasks humans can solve alone compared with help from AIs. This allows to measure and observe whether humans punch above their weight classes after having collaborated with AIs.
* The two-phase design. I was first a bit skeptical about this, as there is no control group. However, after having thought about it, I liked it quite a lot. Because of the reliance on the ELO-score, the same humans that collaborated are the control group when they performed tasks without access to the AI. Also, the fact that humans are not allowed to rely on the output directly, as they cannot transfer notes or code proposed by the AI, strongly suggests learning. However, it could also be thought of as a test of memory. Come to think of it, how could one ensure that participants did not just remember the solution that was proposed by the LLM?
* I like the conceptual framework for knowledge transfer. This is a good way of framing knowledge transfer.
* The incentive mechanisms are nice, especially that the tasks prepare the participants to relevant jobs. Well done!
* I like Figure 4. It contains quite a lot of information but structures it well.
## WEAKNESSES
This paper juggles two very different research questions. One is related to knowledge transfer from AI to human, which the two-phase experimental design seeks to capture. The other relates to whether LLMs are good teachers, which the title alludes to: “when models know more than they can explain”. It is also mentioned as a crucial assumption on line 18. The problem is not that it seeks to answer these questions, but that the design is framed to capture one and the results are analyzed to answer the other. Let me explain why I think this is the case.

Figure 1 (left) plots model performance versus Human-AI collaboration performance and concludes that models improve human-AI collaboration. I do not understand how one could draw this conclusion from the figure. In order to enable us to draw this conclusion, I would like to see the plot of human performance vs collaboration performance instead. Then, one could say something about whether AI-collaboration increases human performance, and one could conclude that humans learn from collaborating from with the LLM.

What I see instead is that: 1) for some models, such as GPT-4o both math and coding, the human AI collaboration has much higher performance than the models, which indicates that human ability is the source of the collaboration performance, 2) for some models, such as Gem-2.5, the value of collaboration is negative as humans are not able to learn from LLMs; it would be better to have the LLM perform the task.  The second point illustrates the title “when models that know more than they can explain”. Some models are not able to transfer their performance to humans in a collaboration to increase the collaboration performance beyond the solo model performance (point 2 above). Thus, in theory at least, it would be better to have the LLM do the task.

On line 44 it is stated that “figure 1 illustrates the participants ability to integrate model-generated reasoning into their own reasoning”, but I do not understand how figure 1 illustrates the participants ability to integrate model-generated reasoning into their own reasoning. Could you please elaborate why you draw this conclusion? The same goes for line 45: I do not understand how one could say that it is the models that enable collaboration based on their solo performance. How can you be sure that it is the AI that enables good performance in human-AI collaborations when collaboration performance is better than model performance and that the performance would not be as good if humans alone performed the task? I do not see the causal relationship.

I think this paper would gain from clearly specifying that the research is limited to tasks and workflows in which humans maintain the primary agency, as is mentioned briefly on line 110.

Although I am not aware of such work, as this is not my area of research, it would surprise me if someone in pedagogy or learning sciences have not tried to quantify the transfer of learning between humans. Even if no such related work exists, I still miss references from outside of AI, especially as this research clearly is interdisciplinary. Hence, I would encourage the authors to search for related work from other related fields.

The section on results lacks the most important result, namely a comparison between Human performance and Collaborative performance. This is a severe omittance.

I like the discussion on subjective preference vs model preference. This is very well explained.

## MINOR COMMENTS
* Line18: Does the rapid progress hide a crucial assumption? I do not see this. Capability of performance of a task does not (necessarily) correlate with the ability to teach. This idea is so strong that there is even a saying (not very nice to teachers though): “Those who can, do. Those who cannot, teach”. I would rephrase that sentence.
* Line 23: I think it is safe to say the goal of all human-AI collaborative workflow is not to outsource thinking to AI. Why have a human in the loop then? Would it be a collaborative workflow then? In human-AI workflows, humans are responsible for the decision or outcome of the workflow, so having the AI do “the thinking” without human understanding would be the same as giving the responsibility to the AI.
* Line 71: “…, our work extends beyond immediate task outcomes to systematically measure reasoning transfer.” This is very nice, but I think you should outline how this is done earlier in the paper and the abstract. Readers are very interested in how this is done, so outlining it early is important.
* Figure 2: The right part on effective projections does not help explain the concept as well as the text does. Hence, it is redundant. If it contains more information than the text, then this must be explained better. If not, it will be better to remove the right part of Figure 2 (or change it to something else).
* I see how the concept of effective projections could be a valuable concept. However, I do not see that effective projections is a concept that this paper digs deep into. The experiments do not evaluate this concept, nor is it used for explanations. Hence, I think it can be removed without any problems, as it does not help the reader understand the research described in the paper.
* Line 127: Can’t => cannot
* Line 129: “This enables us to isolate and measure knowledge transfer from AI to humans.” While I could agree that solving the task is different from discussing it and that many unexpected issues appear when one tries to solve a task practically and that rote learning would not help, I do not understand how the two-step protocol isolate and measure knowledge transfer from AI to humans. How can you be sure that the AI transfer any knowledge to humans? How can we be sure that the humans were not better off without the AI? Would not this require us to have a benchmark of the human capability before doing the task? Would not this require solo performance for humans (ELO score) or a control group?
* Line 165: I do not understand how contrasting the top 25% of problems solved by the model with human skill level avoids any biases. Could you please elaborate?
* How did you end up with the specific thresholds (+/- 220 ELO score for code problems and +/- 0.25 for math problems)?
* On line 181 it is stated that subjective human preferences make it hard to evaluating human-AI collaboration. How come? The design with ELO-scores and testing whether human-AI collaborations perform better than humans alone would solve this, would it not? It measures whether collaboration works without worrying about preferences.
* Line 184: Should there be a reference to the Bradley-Terry model maybe? I like the way of doing this. The model is fine, just cite it.
* 185: Comparing collaboration with model performance is my main gripe with this paper.
* Table 1: What does HTM, HMT, and MHT mean? Please explain.
* Skill hierarchies are not defined. I would think of it like: “you must learn to crawl before you can walk, and you must learn to walk before you can run. This is a hierarchy of skills: crawl > walk > run. It seems to me (if I try to understand this from the caption of Table 1) that with skill hierarchy you mean a hierarchy of skills between humans, tasks and AIs. Is this correct? How is this a skill tree?
* Line 207: The main results can be found in Figure 1 and 2. Do you mean 1 and 4?
* Do you need the extra precision from 4.90 instead of just 4.9?
* Line 211: I have a problem with this whole paragraph. Little makes sense to me here.
* Line 216: “GPT-4o showed strong improvement in math tasks”. How could GPT-4o improve? Is there a training of GPT-4o that used these results? Or do you mean that collaboration performance was much better than the GPT-4o model performance? What can we learn from this? I do not think we can learn much.
* Line 235: Please motivate why you prefer to examine models that helped solve more problems.
* Line 262: Regarding minimal impact of LLM familiarity. I am not sure I follow your hypothesis. Could it be that the users did not actually learn much from collaboration? Have you controlled for this?
* Line 296: Regarding coding tasks that benefitting from better alignment between the procedural nature of the task and the model. How does this related to what is stated earlier regarding math models on line 230? On line 230, you attribute the issues to formal notation, dense symbolic expressions and proof-based explanations. I would assume, although hard to understand, math explanations would make sense procedurally. Is this correct or did I misunderstand something? None of the arguments on line 230 substantiate the argument on line 296 regarding the procedural nature. Please explain.
* Line 329: I do not understand which results substantiate the statement “As models grow more capable, …”. This sentence does maybe not belong in the conclusion?
## REFERENCES

Kosmyna, N., Hauptmann, E., Yuan, Y. T., Situ, J., Liao, X. H., Beresnitzky, A. V., ... & Maes, P. (2025). Your brain on chatgpt: Accumulation of cognitive debt when using an ai assistant for essay writing task. arXiv preprint arXiv:2506.08872.

---

> ### Author Rebuttal · Authors · 2025-07-30
>
> Thank you for this very thorough and thoughtful review. We are *very* happy that you recognize the significance and the high potential of our work. In summary, we believe that the main issues you have with our work can be fully addressed by providing clarification on the “fixed elo” aspect of our question assignment process.
>
> ### **Addressing Core Issues:**
> > Therefore: the problem is not that it seeks to answer these questions, but that the design is framed to capture one and the results are analyzed to answer the other.
>
> > Figure 1 (left)... I do not understand how one could draw this conclusion from the figure.
>
> > How can you be sure that it is the AI that enables good performance in human-AI collaborations?
>
>
> We completely understand your concern, but we do not believe this to be the case. Our experiment is designed so that every participant tackles problems they have only a constant, well‑quantified chance of solving unaided. During each session we select a task whose difficulty is, on average, a *fixed* amount above the participant’s own ELO: specifically, code $+300 \pm 100$ Elo, math $+1 \pm 0.25$ Elo.  Because task and participant Elos are calibrated with a Bradley-Terry model, the Elo difference immediately gives an estimated solo‑success probability via the logistic Bradley–Terry curve. Empirically, this yields expected unaided accuracies of  ~14 % for code and ~8 % for math. Because the solo‑success probability (on average) is a fixed, known constant for every participant, subtracting it would just shift every point on Figure 1 downward by the same amount and leave all the relative distances intact. In other words, the collaboration accuracy already tells us how much ground the Human + AI pair gains over that baseline. For instance, if the team achieves 50 % accuracy in code, the model is delivering an extra $50-14=36$% of performance, exactly the knowledge transfer we care about. So an explicit “collaboration minus baseline” plot would deliver similar conclusions.
>
> To be certain that this calibration was valid (i.e., that participants truly felt the tasks were just out of reach), we ran a preliminary cohort in which we asked, post‑interaction, whether they believed they could have solved the task unaided. The answers supported our Elo gaps, and we locked them in before running the full study. We also ensured balanced skill distributions across treatment groups (details in Section 4.2/4.3 and Appendix B), so any remaining variance comes from the models, not from uneven human ability. In short, the fixed‑ELO protocol lets us treat collaboration accuracy as a quantitative proxy for knowledge transferred from model to human. This perspective was very helpful, and we have updated Section 4 to make this a lot clearer in our paper by fully detailing this design decision.
>
> > What I see instead is that: 1) for some models, such as GPT-4o both math and coding, the human AI collaboration has much higher performance than the models, which indicates that human ability is the source of the collaboration performance, 2) for some models, such as Gem-2.5, the value of collaboration is negative as humans are not able to learn from LLMs
>
> Hopefully this is also addressed in the first response!
>
> > Come to think of it, how could one ensure that participants did not just remember the solution that was proposed by the LLM?
>
> This is a great point, and is explained in Section 3.1. Essentially, we purposely choose automatically verifiable domains where there is an “ideation” phase and an “implementation” phase. Much prior work in cognitive transfer and learning has shown that transforming conceptual instructions into executable solutions requires far more than rote memory [4, 5]. To describe this very concretely, take this canonical solution of a sample coding problem provided by GPT-4o:
>
>
> To solve this problem, we simulate the process of dropping each square onto the X-axis, keeping track of where each square lands and updating the current maximum stack height accordingly.
> Strategy:
> Iterate through the list of squares.
>
>
> For each square, check all previously dropped squares to see if they overlap horizontally.
>
>
> If they do, take the maximum height from those squares as the base height.
>
>
> The new height is then this base height + side length of the current square.
>
>
> Track the maximum height so far after each drop
>
> Even if participants were to memorize the model’s explanations verbatim, successfully implementing these ideas in code demands a deeper understanding. Translating natural language descriptions into working solutions requires selecting appropriate data structures, correctly implementing the algorithm, and efficiently structuring code: all of which go beyond rote recall. This is precisely why we explicitly disallow direct code generation (or symbolic computation in the case of math) as stated in section 4.1; such outputs could be reused without comprehension, undermining the goal of evaluating genuine knowledge transfer. Moreover, the act of debugging (in programming) or diagnosing incorrect reasoning (in math) serves as an additional filter for understanding. Without grasping the intent and mechanics of a proposed solution, participants cannot meaningfully adapt or correct it.
> > I think this paper would gain from clearly specifying that the research is limited to tasks and workflows in which humans maintain the primary agency, as is mentioned briefly on line 110.
>
>
> Absolutely. Although we do mention it, we will make this clearer as a limitation in the final version.
>
> > The section on results lacks the most important result, namely a comparison between Human performance and Collaborative performance.
>
> In a similar vein to the first response, this information is fully encapsulated in the model performance v collaboration performance comparison.
>
> > it would surprise me if someone in pedagogy or learning sciences have not tried to quantify the transfer of learning between humans.
>
> This is a wonderful point. After doing some further review, we were able to find some additional related papers [1, 2, 3], and we greatly appreciate this pointer. We have added this to the revision of the paper!
>
> ### **Addressing Minor Issues:**
>
> > How did you end up with the specific thresholds (+/- 220 ELO score for code problems and +/- 0.25 for math problems)?
>
> We calibrated this increase on a test cohort. As mentioned in the paper in Section 4.2, we wanted to ensure that these problems represented problems participants could not solve alone, but still had some potential of solving with assistance. This is a detail that we have expanded upon in the paper, as it provides a lot of context.
>
> > On line 181 it is stated that subjective human preferences make it hard to evaluating human-AI collaboration. How come?
>
> This can be clearer for sure. What we mean is that a generalized metric for human-AI collaboration is difficult since it would need to adequately factor in subjective preferences and objective helpfulness. Ie, a very helpful teacher that forces you to do practice problems before solving the core problem may not necessarily be liked.
>
> > Table 1: What does HTM, HMT, and MHT mean? Please explain.
>
> Thank you for bringing this up. This was a poor design decision we made due to space constraints. Essentially it refers to the relative ELO ratings of model, task, and human. (HTM = Human < Task < Model, HMT = Human < Model < Task… etc.).
>
> > Regarding minimal impact of LLM familiarity. I am not sure I follow your hypothesis. Could it be that the users did not actually learn much from collaboration? Have you controlled for this?
>
> This means that our participants’ self-reported prior experience with LLMs did not impact the collaboration performance in a statistically significant fashion. We controlled and tested for potential confounders and covariates as much as possible (detailed in Section 5), and one possible confounder we thought of was whether a human with more experience prompting and interacting with LLMs might give them an edge in solving more problems, no matter which model they were interacting with. Our results show that this was not the case: which is interesting in itself!
>
> Thank you again for your thoroughly detailed review: reading through your understanding of the paper is invaluable to helping us understand what is not clear from a new perspective: especially your thoughts on Figure 1. We have already made many edits in response to your concerns. We believe strongly in our work, so we look forward to further conversation to refine our work and clear up any additional questions!
>
> References:
>
> [1]  S. Hajian, “Transfer of Learning and Teaching: A Review of Transfer Theories and Effective Instructional Practices,” IAFOR J. Educ., vol. 7, no. 1, pp. 93–111, 2019.
>
> [2]  K. Watanabe et al., “Accelerating Knowledge Transfer by Sensing and Actuating Social‑Cognitive States,” in Proc. UbiComp/ISWC ’23 Adjunct, 2023, pp. 258–262.
>
> [3]  O. Elnaggar and R. Arelhi, “Quantification of Knowledge Exchange Within Classrooms: An AI‑Based Approach,” in Eur. Conf. Educ. 2021 Proc., 2021, pp. 221–231.
>
> [4]  K. R. Koedinger and I. Roll, “Learning to Think: Cognitive Mechanisms of Knowledge Transfer,” in *The Oxford Handbook of Thinking and Reasoning*, K. J. Holyoak and R. G. Morrison, Eds. Oxford: Oxford Univ. Press, 2012, pp. 699–715.
>
> [5]  O. Chen, E. Retnowati, B. K. Y. Chan, and S. Kalyuga, “The Effect of Worked Examples on Learning Solution Steps and Knowledge Transfer,” Educ. Psychol., vol. 43, no. 8, pp. 914–928, 2023.

---

> > ### Comment · Reviewer_wnSo · 2025-08-06
> > **Thank you for a detailed response to my comments and questions**
> >
> > My concerns are stilled. I really like your research, and I hope it will be accepted for presentation.

---

> > > ### Author Response · Authors · 2025-08-06
> > >
> > > We are glad to hear that our response has addressed all your concerns! Thank you very much for your detailed review, invaluable suggestions, and making the paper review process a valuable and constructive experience for us.

---

### Note · Authors · 2025-08-12

We thank all reviewers for their careful, constructive feedback. Below we aggregate the key concerns, describe the changes already implemented, and reiterate why our study is essential.
### **Significance**
As LLMs are pushed into tutoring products, study modes, and high‑stakes workplaces, many implicitly assume that scaling model reasoning will *automatically* make models better at transferring their reasoning to humans. No prior work (to our knowledge) quantitatively tests this claim. Our study supplies the first framework to measure such *knowledge transfer*, made possible by an Elo‑calibrated Bradley-Terry design that halves per‑participant sample needs on 118 users across 8 frontier models. By exposing cases where top‑performing models fail to transfer and others where weaker models excel, we surface a critical blind spot in current evaluation and provide an actionable framework for alignment. Section 1 now foregrounds this contribution and clarifies that our framework pertains to tasks with automatic grading; broader domains are flagged as future work.
### **Key reviewer concerns & our actions**
1. *Calibration & baseline clarification:*  Tasks are fixed at +300 ± 100 Elo (code) / +1 ± 0.25 Elo (math) per participant, implying ≈14 % / 8 % solo‑success odds by BT model. Figure 1 now marks this “solo‑chance”; we now contextualize chosen hyperparameters in pilot studies.

2. *Human–human control: Claim:* collaboration benefits could be due to generic effects. *Response:* Any partner‑presence boost adds the same constant to all eight models, leaving our results unchanged. Also, mixed‑effects regression finds model identity highly significant (p < 0.001) after covariate controls. Human-human setting is interesting, but out-of-scope for this work.

3. *Closed‑book phase & note ban: Claim:* prohibiting notes measures memory, not learning. *Response:* This decision is grounded in long-standing education research on closed vs open book testing. Pilot sessions with notes showed copy‑paste behaviour and hid learning; Sec 3.1 now contextualizes this design properly.

4. *Prompting & temperature: Claim:* Uniform prompt and T = 0.7 may handicap some models. *Response:* A fixed temperature + prompt removes decoding confounds and approximates provider defaults; Section 4 now explains this clearer.

5. *Other Clarity gaps:* HTM/HMT/MHT now defined in Table 1, Fig. 2 trimmed, and learning‑science citations added.

Thanks again to all reviewers!!

---

### Decision · Program_Chairs · 2025-09-17

**Decision:**

Accept (poster)

**Comment:**

This paper introduces a novel framework for quantifying knowledge transfer from LLMs to humans through a two-phase collaborative study with 118 participants across coding and math tasks. The key finding is that model benchmark performance correlates with but doesn't fully predict collaborative success, suggesting knowledge transfer is a distinct capability requiring separate evaluation. The paper's primary strengths include its well-executed experimental design that cleanly isolates knowledge transfer effects, the significant practical relevance of measuring AI teaching effectiveness, and robust methodology with Elo-calibrated task difficulty and mixed-effects analysis controlling for confounders. However, the work has notable limitations: the evaluation is confined to STEM domains with automatic verification, lacks human-human collaboration baselines to isolate AI-specific effects, and imposes potentially artificial constraints like prohibiting note-taking and pseudocode that may reduce ecological validity. The experimental choices around uniform temperature settings and different submission limits across tasks also lack sufficient justification. Despite these methodological concerns, the paper addresses a crucial gap in understanding human-AI collaboration and provides the first systematic framework for measuring knowledge transfer effectiveness.

The post-rebuttal discussion revealed persistent methodological disagreements, particularly from reviewer aoxv who maintained concerns about the experimental design despite acknowledging the technical contributions. The authors effectively addressed most concerns by clarifying their fixed-Elo calibration approach, which treats collaboration accuracy as a direct measure of knowledge transfer since all participants face tasks with known baseline success probabilities. Reviewers wnSo and M7Bb were satisfied by these explanations and the additional knowledge-transfer gain data provided, with wnSo updating from initial concerns to acceptance. Reviewer zabu remained supportive throughout, and reviewer qEBD maintained their borderline accept position. The key remaining contention centers on whether the study design adequately captures "collaboration" versus "learning," with aoxv arguing that the closed-book constraints better measure memory than collaborative knowledge transfer. Balancing the paper's novel contribution and robust execution against these methodological limitations, the work merits acceptance as it establishes an important new research direction with actionable insights for AI alignment, though future work should address the ecological validity concerns raised.